# Bridging Central and Local Differential Privacy in Data Acquisition Mechanisms

**Alireza Fallah**
EECS Department
Massachusetts Institute of Technology
afallah@mit.edu

**Ali Makhdoumi**
Fuqua School of Business
Duke University
ali.makhdoumi@duke.edu

**Azarakhsh Malekian**
Rotman School of Management
University of Toronto
azarakhsh.malekian@rotman.utoronto.ca

**Asuman Ozdaglar**
EECS Department
Massachusetts Institute of Technology
asuman@mit.edu

## Abstract

We study the design of optimal Bayesian data acquisition mechanisms for a platform interested in estimating the mean of a distribution by collecting data from privacy-conscious users. In our setting, users have heterogeneous sensitivities for two types of privacy losses corresponding to local and central differential privacy measures. The local privacy loss is due to the leakage of a user's information when she shares her data with the platform, and the central privacy loss is due to the released estimate by the platform to the public. The users share their data in exchange for a payment (e.g., through monetary transfers or services) that compensates for their privacy losses. The platform knows the distribution of privacy sensitivities but not their realizations, and must design a mechanism to solicit their preferences and then deliver both local and central privacy guarantees while minimizing the estimation error plus the expected payment to users. We first establish minimax lower bounds for the estimation error, given a vector of privacy guarantees, and show that a linear estimator is (near) optimal. We then turn to our main goal: designing an optimal data acquisition mechanism. We establish that the design of such mechanisms in a Bayesian setting (where the platform knows the distribution of users' sensitivities and not their realizations) can be cast as a nonconvex optimization problem. Additionally, for the class of linear estimators, we prove that finding the optimal mechanism admits a Polynomial Time Approximation Scheme.

## 1 Introduction

Users' personal data are currently being utilized for personalized advertising, medical trials, targeted advertising, and recommendation systems, among others. The transaction of individual data is set to grow exponentially in the coming years, with more widespread applications of artificial intelligence (AI) and machine learning techniques. Even though it is widely accepted that users need to own their data (see, e.g., Posner and Weyl [2019], Kushmaro [2021], and WILL.I.AM [2019]), the impact of different market architectures on the design and operation of data markets are not clear: some users prefer to protect their own raw data while others expect companies to protect their data proactively (see, e.g., GDMA [2018]). In this paper, we consider the design of data acquisition mechanisms when users have heterogeneous privacy concerns and ask the following question:

36th Conference on Neural Information Processing Systems (NeurIPS 2022).

*What is the optimal data acquisition mechanism when users have heterogeneous privacy concerns regarding access to their raw data and the outcome of the platform's processing?*

We use differential privacy to measure the two types of privacy losses. Informally, an estimator is called differentially private if its distribution over outputs is insensitive to the changes in a user's data.

In particular, we consider a platform whose goal is to estimate an underlying parameter of interest by collecting data from a set of users $\mathcal{N} = \{1, \ldots, n\}$ who own a noisy version of the underlying parameter. For instance, consider a medical trial in which a hospital wants to collect users' data to estimate the efficacy of a drug. Each user has two types of privacy concerns: (i) local privacy concern that captures how much information their shared data reveal about their raw data, and (ii) central privacy concern that captures how much information the platform's output reveal about their raw data. We adopt local and central Rényi differential privacy to measure these two types of privacy losses. The reason for choosing Rényi differential privacy is twofold. First, our framework can cover a wide range of information measures by varying the Rényi divergence parameter. Second, it can be achieved by a Gaussian mechanism, which simplifies our analysis while capturing the main tradeoffs in the design of two-part data acquisition mechanisms.

Before formulating the platform's data acquisition problem, we derive optimal estimators for a given vector of heterogeneous local privacy loss levels. In particular, we establish a minimax lower bound for the estimation error and prove that first privatizing users' data by adding a properly designed Gaussian noise to them and then using a properly designed weighted average of these privatized data points achieves this lower bound. This result motivates us to consider the design of the optimal data acquisition mechanism for the class of linear estimators.

We then turn to our mechanism design problem. Each user has a heterogeneous preference regarding the importance of the above two privacy concerns. For instance, if a user fully trusts the platform, then the first type of concern lessens, and the main concern would be about the information revealed from the platform's estimate. On the other hand, if a user does not trust the platform at all, the first type of concern would be more than the second one. We model such a setting by assuming each user $i$ has a privacy sensitivity $c_i \in [0, 1]$ that determines the relative weight she puts on the local privacy concern ($1 - c_i$ is the weight she puts on the central privacy concern). The utility of user $i$ is the payment she receives from the platform (in exchange for sharing her data), minus $c_i$ times her local privacy loss, and again, minus $1 - c_i$ times her central privacy loss. The platform does not know the value of $c_i$ and (knowing its distribution) must design a (Bayesian) data acquisition mechanism to elicit the true privacy sensitivities (that guide the optimal choice of local and central privacy losses delivered to each user) and optimize its objective.

In particular, the platform designs a *two-part data acquisition mechanism* that comprises a payment scheme, a local privacy guarantee, and a central privacy guarantee as a function of the reported privacy sensitivity of users. The platform's goal is to minimize the sum of the mean estimation error and the expected total payment to users while satisfying the *incentive compatibility* and *individual rationality* constraints. Incentive compatibility ensures that users have no incentive to misreport their privacy sensitivity, and individual rationality ensures that the payment to users (and the delivered privacy guarantees) are such that users are willing to share their data with the platform.

The platform's problem is a functional optimization over three functions of the reported privacy sensitivities: payments, local privacy guarantees, and central privacy guarantees. We first find the payment function in terms of the local and central privacy guarantees by using the incentive compatibility and individual rationality constraints. This reduces the space of the platform's decision variables. We then focus on the Gaussian mechanisms and linear estimators (motivated by our minimax optimality result) and show that the platform's problem can be cast as an optimization problem that minimizes a non-convex objective (which depends on the *virtual cost* of users) for any reported vector of privacy sensitivities. This reformulation significantly reduces the space of decision variables that the platform needs to optimize. However, it still involves solving a non-convex optimization problem. We further use the structural properties of this non-convex optimization and use duality theory to develop a polynomial time algorithm to approximate the platform's problem. More precisely, we prove that the design of the optimal two-part data acquisition mechanism admits a Polynomial Time Approximation Scheme (PTAS).

The contribution of our work is threefold. First, we develop a minimax lower bound when users have heterogeneous local privacy losses and establish that a linear estimator (approximately) achieves this bound. Second, we develop a modeling framework for data acquisition mechanisms when users have heterogeneous concerns for both local and central privacy losses. Third, for any estimator and mechanism to deliver privacy guarantees, we characterize the design of the optimal two-part data acquisition mechanism as the solution to a point-wise optimization problem. Additionally, for the class of Gaussian mechanisms to deliver privacy guarantees and linear estimators, we develop an algorithm to approximately find the optimal data acquisition mechanism (despite the fact that the corresponding optimization is non-convex).

**Related literature:** Our paper relates to the literature on optimal data acquisition from privacy concerned users. There is a large body of work that use differential privacy to measure the privacy loss of users Ghosh and Roth [2011], Nissim et al. [2012], Nissim et al. [2014]. One of the earliest papers in the literature is Ghosh and Roth [2011], which study the design of mechanisms for collecting users' data when users incur some privacy cost from sharing their data. More specifically, Ghosh and Roth [2011] consider binary data (bit) with the platform's goal being to estimate the sum of user's data by using a differentially private and dominant strategy truthful mechanism. They study both the case when the user data and privacy parameter are independent (similar to our paper) and when they are correlated. In the independent case, they propose a mechanism that delivers a single privacy level to all users (as opposed to our setting that delivers heterogeneous privacy levels). For the correlated case, they prove an impossibility result for the existence of a truthful and individually rational mechanism.

Several works build on Ghosh and Roth [2011], extending it to take it or leave it offers Ligett and Roth [2012], strengthening the impossibility results Nissim et al. [2014], and studying the open question posed by Ghosh and Roth [2011] on whether a model with distributional assumption on users' costs and Bayesian mechanism design approach could be used to develop the optimal mechanism for collecting data with privacy guarantees (see, e.g., Liao et al. [2021] and Fallah et al. [2022]). In particular, Roth and Schoenebeck [2012], Chen et al. [2018], and Chen and Zheng [2019] tackle this problem by developing a randomized mechanism in which user's data is randomly included in the final estimator where the inclusion probability depends on the reported privacy costs of the users (as opposed to our setting in which the payments and privacy guarantees depend on the reported privacy sensitivity of all users). These papers do not use differential privacy to model privacy costs and instead use a menu of probability-price pairs to tune the privacy loss and the payment to each user (see also Pai and Roth [2013] for a survey). Similar to the above paper, we consider a setting in which the platform can verify the data of users. For instance, in the context of medical trials, this means that the users decide whether to participate in the medical trial and cannot change the samples they share. A different stream of the literature explores settings in which users can misreport their information Perote and Perote-Pena [2003], Dekel et al. [2010], Meir et al. [2012], Ghosh et al. [2014], Cai et al. [2015], Liu and Chen [2016, 2017].

Our paper differs from these works in three main ways. First, we assume prior information on user privacy sensitivities and focus on characterizing the optimal Bayesian incentive compatible mechanism. Second, we model a setting in which users have both local and central privacy concerns and explore the different privacy guarantees of these two types delivered by an optimal mechanism. Third, we assume that user data are drawn from the same underlying distribution that allows the platform to put differing weights on the data of users depending on their privacy sensitivity, leading to different privacy levels for participating users.

Finally, our paper relates to the literature on differential privacy. Pioneered by the seminal work of Dwork et al. [2006a,b], differential privacy has emerged as a prevalent framework for characterizing the privacy leakage of data oriented algorithms. More specifically, our paper is related to the private mean estimation considered by Duchi et al. [2013], Barber and Duchi [2014], Karwa and Vadhan [2017], Asoodeh et al. [2021], Kamath et al. [2019, 2020], Cummings et al. [2021], and Acharya et al. [2021]. Additionally, our paper relates to the stream of differential privacy literature that studies Rényi differential privacy (RDP) introduced by Bun and Steinke [2016] and Mironov [2017].

## 2 Problem Formulation

We consider a platform interested in estimating a parameter $\theta \in \mathbb{R}$ by collecting data of $n$ users, indexed by $\mathcal{N} = \{1, \cdots, n\}$. For any $i \in \mathcal{N}$, we denote user $i$'s data by $X_i \in \mathcal{X}$ and we assume $X_i$

is given by $X_i = \theta + Z_i$ where $Z_1, \cdots, Z_n$ are independent and identically distributed zero-mean random variables with variance VAR. To simplify the exposition, we further assume $|Z_i| \leq 1/2$ for any $i \in \mathcal{N}$.[1] Throughout, we use lower case letters to denote the realization of random variables. The platform's goal is to minimize the estimate's error by collecting data from privacy-concerned users. Therefore, the platform needs to incentivize them to share their data.

## 2.1 Local and central privacy losses

Before formalizing the utilities/objectives of the platform and the users, we define the notions of privacy losses that we adopt in this paper. In particular, we consider two different types of privacy losses that users suffer from. The first one is the privacy loss of a user when she shares her data (only) with the platform, and the second one is the privacy loss through the released estimate (to the public) by the platform. Depending on how different users trust the platform, they might care differently about these two privacy losses. For instance, if a user fully trusts the platform, then her main privacy concern would be the second one, while a user who does not trust the platform at all would be more concerned with the first one as the public only observes the aggregated estimate, as opposed to the platform which observes each user's (shared) data separately.

We use the differential privacy framework to quantify these privacy losses. Since differential privacy was introduced by Dwork et al., several variants of it have been also proposed. In particular, a popular one in the machine learning literature is Rényi differential privacy (RDP), introduced by Mironov [2017], which we also adopt in this paper. Let us first recall the definition of Rényi divergence.

**Definition 1.** *Let $P$ and $Q$ be two distributions over $\mathbb{R}$ with densities $p$ and $q$. For any $\alpha \in (1, \infty]$, the Rényi $\alpha$-divergence between $P$ and $Q$ is denoted by $D_\alpha(P||Q)$ and is given by*

$$D_\alpha(P||Q) := \frac{1}{\alpha - 1} \log \int \left( \frac{p(x)}{q(x)} \right)^\alpha q(x) dx.$$

*For two random variables $X$ and $Y$, $D_\alpha(X||Y)$ denotes the $\alpha$-divergence between their distributions.*

We next define two notions of differential privacy, known as central and local, to capture the two aforementioned types of privacy losses. Local differential privacy corresponds to the privacy loss of a user when she shares her data with the platform through a randomized mapping, known as a *channel*.

**Definition 2.** *Let $\varepsilon \geq 0$ and $\alpha \in (1, \infty]$. A randomized channel $\mathcal{C} : \mathcal{X} \to \mathbb{R}$ is locally $(\varepsilon, \alpha)$-Rényi (differentially) private if for any $x, x' \in \mathcal{X}$,*

$$D_\alpha(\mathcal{C}(x)||\mathcal{C}(x')) \leq \varepsilon.$$

Central differential privacy corresponds to the other privacy loss mentioned above. It bounds the change in the distribution of the platform's output, i.e., the released estimate, by changing one user's data. We next provide the formal definition.

**Definition 3.** *Let $\boldsymbol{\varepsilon} = (\varepsilon_i)_{i=1}^n \in \mathbb{R}_+^n$ and $\alpha \in (1, \infty]$. A randomized algorithm $\mathcal{A} : \mathcal{X}^n \to \mathbb{R}$ is $(\boldsymbol{\varepsilon}, \alpha)$-Rényi (differentially) private if for any two datasets $x_{1:n}, x'_{1:n} \in \mathcal{X}^n$ that only differ in $i$-th coordinate (data of user $i$),*

$$D_\alpha(\mathcal{A}(x_{1:n})||\mathcal{A}(x'_{1:n})) \leq \varepsilon_i.$$

The customary approach to guarantee RDP is Gaussian mechanism in which a properly tuned zero-mean Gaussian noise is added to fulfill the required condition. The following lemma, adapted from Mironov [2017], allows us to characterize the Gaussian noise's variance for a privacy loss level.

**Lemma 1.** *For a function $f : \mathcal{X}^n \to \mathbb{R}$, we define its sensitivity with respect to $i$-th coordinate as*

$$L_i(f) := \sup \left\{ |f(x_{1:n}) - f(x'_{1:n})| \; : \; \text{for all } x_{1:n} \text{ and } x'_{1:n} \text{ differing only at } i\text{-th coordinate} \right\}.$$

*For any $\alpha \in (1, \infty]$, $\mathcal{A}(x_{1:n}) = f(x_{1:n}) + W$ with $W \sim \mathcal{N}(0, \sigma^2)$ is $\left( (\frac{\alpha L_i(f)^2}{2\sigma^2})_{i=1}^n, \alpha \right)$-RDP.*

For a given vector of local privacy losses $(\varepsilon_1^{(l)}, \ldots, \varepsilon_n^{(l)})$, a natural way to privately estimate the mean is by using a linear estimator with Gaussian mechanism which is given by

$$\hat{\theta}(x_1, \ldots, x_n) := \sum_{i=1}^n w_i \hat{x}_i \text{ where } \sum_{i=1}^n w_i = 1 \text{ and } \hat{x}_i = x_i + \mathcal{N}(0, \frac{\alpha}{2\varepsilon_i^{(l)}}) \text{ for all } i \in \mathcal{N}. \quad (1)$$

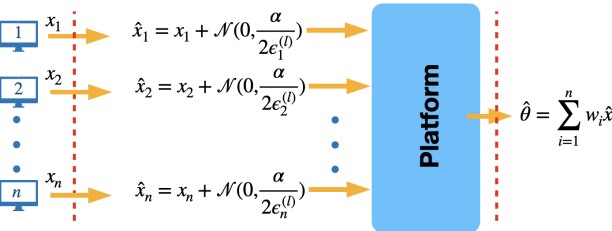

Figure 1: The interaction between the users and the platform in the two-part private data acquisition.

Figure 1 depicts this estimator. In particular, using Lemma 1, the local privacy delivered to user $i \in \mathcal{N}$ is $\varepsilon_i^{(l)}$ and the central privacy delivered to user $i \in \mathcal{N}$ is $\varepsilon_i^{(c)} = w_i^2 / \sum_{j=1}^n w_j^2 / \varepsilon_j^{(l)}$. In the next section, we focus on linear estimators in designing the optimal private data acquisition mechanism. We motivate such specification by showing that, for a given vector of local privacy losses $(\varepsilon_i^{(l)})_{i=1}^n$, a linear estimator is *optimal* with respect to mean square error. To formalize this statement, we first need to define the *minimax* estimation error as the notion of optimality. Let $\mathcal{P}$ be a class of distributions over $\mathcal{X}$. For any $P \in \mathcal{P}$, we denote its mean by $\theta(P)$. A $(\varepsilon_i^{(l)})_{i=1}^n$-locally RDP estimator can be cast as $\hat{\theta}((\mathcal{C}_i(x_i))_{i=1}^n)$, where $\mathcal{C}_i(.)$ is the randomized channel corresponding to user $i$. Let $\mathcal{Q}((\varepsilon_i^{(l)})_{i=1}^n)$ be the class of such $(\varepsilon_i^{(l)})_{i=1}^n$-locally RDP estimators. The minimax estimation error is

$$\mathcal{L}(\mathcal{P}, \mathcal{Q}, (\varepsilon_i^{(l)})_{i=1}^n) := \inf_{\hat{\theta}, \{\mathcal{C}_i\}_{i=1}^n \in \mathcal{Q}((\varepsilon_i^{(l)})_{i=1}^n)} \sup_{P \in \mathcal{P}} \mathbb{E}_{(X_i \sim P)_{i=1}^n, \hat{\theta}}[|\hat{\theta}((\mathcal{C}_i(X_i))_{i=1}^n) - \theta(P)|^2], \quad (2)$$

where the expectation is taken over both randomness of data and estimator (including private channels). In other words, the optimal estimator is the one that has the lowest worst case error among all estimators that satisfy the privacy requirements. We next state the optimality result.

**Theorem 1.** *Assume $\alpha \geq 2$ and $\varepsilon_i^{(l)} \leq 1$ for all $i$. Let $\mathcal{P}_1$ be the family of distributions over $[-\frac{1}{2}, \frac{1}{2}]$ and $\mathcal{C}_1, \cdots, \mathcal{C}_n$ be independent channels. Then, there exists a universal constant $c$ such that*

$$\mathcal{L}(\mathcal{P}, \mathcal{Q}, (\varepsilon_i^{(l)})_{i=1}^n) \geq c \min(\frac{1}{\sum_{i=1}^n \varepsilon_i^{(l)}}, 1).$$

*Furthermore, there exists a linear estimator with Gaussian mechanism such that*

$$\mathbb{E}_{(X_i \sim P)_{i=1}^n, \hat{\theta}}[|\hat{\theta}((\mathcal{C}_i(X_i))_{i=1}^n) - \theta(P)|^2] \leq \mathcal{O}(1) \frac{\alpha}{\sum_{i=1}^n \varepsilon_i^{(l)}}.$$

We prove the lower bound by using the Le Cam's method Yu [1997] that reduces the problem of finding lower bounds to a hypothesis testing problem and requires bounding the total variation distance between the estimates when the input data points change. We then find it more convenient to bound the total variation in terms of Hellinger distance and use a series of inequalities to bound it.

## 2.2 Data acquisition mechanism with two-part privacy guarantees

We next describe the utility functions of the users and the platform and then formulate the platform's optimal data acquisition mechanism. As we described earlier, each user suffers from two privacy losses when sharing her data. The first one is a central privacy loss because of the leakage of her information through the platform's output. The second one is a local privacy loss because of the leakage of her information through the raw data that she shares with the platform. Each user has a heterogeneous *privacy sensitivity* for these two types of privacy losses. To model such heterogeneity, for each $i \in \mathcal{N}$, we let $c_i \in [0, 1]$ be her *relative local privacy sensitivity*, representing the relative weight that user $i$ assigns to the (per unit cost of) local privacy loss. We also let $1 - c_i$ be her *relative central privacy sensitivity*, representing the relative weight that user $i$ assigns to the (per unit cost of) central privacy loss. Therefore, $c_i \approx 1$ implies that user $i$ suffers a higher loss of privacy by sharing her raw data with the platform (local privacy loss) compared to her loss from the platform's output

---

[1]This assumption does not have any fundamental impact on the results and is made to simplify the notations.

(central privacy loss). Differently, $c_i \approx 0$ implies that the user suffers a smaller loss of privacy by sharing her raw data with the platform compared to her loss from the platform's output. In what follows, we use the term *privacy sensitivity* instead of relative local privacy sensitivity.

For each $i \in \mathcal{N}$, the privacy sensitivity $c_i$ is independently drawn from a publicly known distribution whose support is $[0, 1]$ with cumulative distribution and probability density functions $F_i(\cdot)$ and $f_i(\cdot)$. We also let $\mathbf{c} = (c_1, \ldots, c_n)$ be the vector of privacy sensitivities. The privacy sensitivity of each user is her private information, i.e., the platform does not know it. This is because individuals have different views regarding how trustworthy the platform is in protecting their raw data.

The platform's objective is to design a mechanism to collect users' data by paying them to compensate for their privacy losses without knowing the privacy sensitivity of users. To introduce the platform's objective formally, we adopt the formalism of Bayesian mechanism design pioneered by Myerson [1981]. More specifically, the platform designs and announces a payment function, a local privacy loss function, and a central privacy loss function that are mappings from the reported privacy sensitivities of users. The users then report their privacy sensitivities (which may or may not be truthful). Based on the payment function, the platform compensates the users (the compensation could be monetary or some free or discounted service provided to the user). Based on the local and central privacy functions, the platform designs randomized channels and randomized estimation algorithms that deliver the guaranteed local and central privacy losses while minimizing the sum of the mean squared error and the total expected payments. Given this interaction, we next formally introduce a data acquisition mechanism with two-part data privacy guarantees.

**Definition 4** (two-part private data acquisition mechanism)**.** We call the tuple $(\hat{\theta}, \boldsymbol{\varepsilon}^{(l)}, \boldsymbol{\varepsilon}^{(c)}, \mathbf{t})$ a *two-part private data acquisition mechanism* where

1. For all $i \in \mathcal{N}$, $\varepsilon_i^{(l)} : \mathbb{R}_+^n \to \mathbb{R}_+$ is a function that maps the vector of privacy sensitivities $\mathbf{c}$ to a local privacy loss for user $i$, $\varepsilon_i^{(l)}(\mathbf{c})$, with $\boldsymbol{\varepsilon}^{(l)}(\cdot) = (\varepsilon_i^{(l)}(\cdot))_{i=1}^n$.

2. For all $i \in \mathcal{N}$, $\varepsilon_i^{(c)} : \mathbb{R}_+^n \to \mathbb{R}_+$ is a function that maps the vector of privacy sensitivities $\mathbf{c}$ to a central privacy loss for user $i$, $\varepsilon_i^{(c)}(\mathbf{c})$, with $\boldsymbol{\varepsilon}^{(c)}(.) = (\varepsilon_i^{(c)}(\cdot))_{i=1}^n$.

3. $\hat{\theta} : \mathcal{X}^n \times \mathbb{R}_+^n \times \mathbb{R}_+^n \to \mathbb{R}$ is a $(\boldsymbol{\varepsilon}^{(c)}(\mathbf{c}), \alpha)$-Rényi differentially private estimator that maps acquired locally $(\varepsilon_i^{(l)}(\mathbf{c}), \alpha)$-Rényi differentially private data of user $i$ for $i \in \mathcal{N}$ to an estimate $\hat{\theta}(\mathbf{x}, \boldsymbol{\varepsilon}^{(l)}(\mathbf{c}), \boldsymbol{\varepsilon}^{(c)}(\mathbf{c}))$.

4. For all $i \in \mathcal{N}$, $t_i : \mathbb{R}_+^n \to \mathbb{R}_+$ is a function that maps the vector of privacy sensitivities $\mathbf{c}$ to a payment for user $i$, $t_i(\mathbf{c})$, with $\boldsymbol{t}(.) = (t_i(\cdot))_{i=1}^n$.

Notice that we have not specified the estimator and the mechanisms that delivers (local and central) Rényi differential privacy. In the rest of this subsection, we introduce the utilities and the platform's problem for a general estimators and mechanisms to deliver differential privacy. Later, we focus on linear estimator and Gaussian mechanisms and explicitly solve the platform's problem.

Each user that participates in a two-part private data acquisition mechanism suffers from both the local and central privacy losses and need to be compensated by the platform. In particular, the utility of user $i$ from participation when her privacy sensitivity is $c_i$ and she reports $c_i'$ is given by

$$u_i(\boldsymbol{\varepsilon}^{(l)}(c_i', \mathbf{c}_{-i}), \boldsymbol{\varepsilon}^{(c)}(c_i', \mathbf{c}_{-i}), \mathbf{t}, \hat{\theta}) = \mathbb{E}_{\mathbf{c}_{-i}}[t_i(\mathbf{c}_{-i}, c_i')) - c_i \varepsilon_i^{(l)}(\mathbf{c}_{-i}, c_i') - (1 - c_i)\varepsilon_i^{(c)}(\mathbf{c}_{-i}, c_i')],$$

where the term $t_i(\mathbf{c}_{-i}, c_i'))$ is the payment from the platform, the term $c_i \varepsilon_i^{(l)}(\mathbf{c}_{-i}, c_i')$ is the relative local privacy sensitivity of the user multiplied by her local privacy loss, and the term $(1 - c_i)\varepsilon_i^{(c)}(\mathbf{c}_{-i}, c_i')$ is her relative central privacy sensitivity multiplied by her central privacy loss. A user $i \in \mathcal{N}$ that does not participate in the mechanism neither compromises her privacy nor gets a compensation. Therefore, the utility of a user who does not participate in the mechanism becomes $0$.

The goal of the platform is to minimize the sum of the mean squared error and the overall payment to users. We let $\gamma \in \mathbb{R}_+$ represents the relative weight of the mean estimation error and the payments in

the platform's objective.[2] Therefore, the platform's objective is

$$\mathbb{E}_{\mathbf{c}}[\gamma \text{MSE}(\varepsilon^{(l)}(\mathbf{c}), \varepsilon^{(c)}(\mathbf{c}), \hat{\theta}) + \sum_{i=1}^{n} t_i(\mathbf{c})],$$

where the first term is the mean square error of estimator $\hat{\theta}$ given reported vector of privacy sensitivity and resulting local and central privacy losses $\varepsilon^{(l)}$ and $\varepsilon^{(c)}$, i.e.,

$$\text{MSE}(\varepsilon^{(l)}(\mathbf{c}), \varepsilon^{(c)}(\mathbf{c}), \hat{\theta}) = \mathbb{E}_{\mathbf{x}}[|\hat{\theta}(\hat{\mathbf{x}}, \varepsilon^{(l)}, \varepsilon^{(c)}) - \theta|^2].$$

Also, each summand of the second term is the compensation that the platform gives to a user to incentivize her to participate and report her privacy sensitivity truthfully.

In Appendix we prove that, similar to the classical mechanism design setting, *revelation principle* holds. This means that there is no loss of generality in focusing on the class of direct incentive compatible mechanisms, meaning the platform's optimization problem can be written as

$$\min_{\varepsilon^{(l)}(\cdot), \varepsilon^{(c)}(\cdot), \mathbf{t}(\cdot)} \quad \mathbb{E}_{\mathbf{c}}[\gamma \text{MSE}(\varepsilon^{(l)}(\mathbf{c}), \varepsilon^{(c)}(\mathbf{c}), \hat{\theta}) + \sum_{i=1}^{n} t_i(\mathbf{c})] \tag{3}$$

$$u_i(\varepsilon^{(l)}(\mathbf{c}), \varepsilon^{(c)}(\mathbf{c}), \mathbf{t}, \hat{\theta}) \geq u_i(\varepsilon^{(l)}(c_i', \mathbf{c}_{-i}), \varepsilon^{(c)}(c_i', \mathbf{c}_{-i}), \mathbf{t}, \hat{\theta}) \tag{4}$$

$$u_i(\varepsilon^{(l)}(\mathbf{c}), \varepsilon^{(c)}(\mathbf{c}), \mathbf{t}, \hat{\theta}) \geq 0 \quad \text{for all } i \in \mathcal{N}, c_i, \tag{5}$$

where the constraints in (4) represent the *incentive compatibility*. These constraints guarantee that that each user $i$ has no incentive to misrepresent her privacy sensitivity when others report truthfully (reporting truthfully is an equilibrium of the game among the users). Also, the constraints in (5) represent *individual rationality*, which ensures that each user receives a non-negative utility from participating in the platform's mechanism and sharing her data.

## 3 From the mechanism design problem to an optimization problem

For a given estimator $\hat{\theta}$, the platform's decision comprises the local and central privacy loss functions $\varepsilon^{(l)}(\cdot)$ and $\varepsilon^{(c)}(\cdot)$ together with the payment functions $\mathbf{t}(\cdot)$. We next show that this problem can be equivalently formulated as an optimization problem over the vector of local privacy losses and central privacy losses (as opposed to functions). In the rest of the paper, we impose the following assumption which is well-known in the mechanism design literature and simplifies the analysis.[3]

**Assumption 1.** *For any user $i \in \mathcal{N}$, the* virtual cost *defined as $\psi_i(c) = c + \frac{F_i(c)}{f_i(c)}$ is increasing in $c$, where $f_i(\cdot)$ and $F_i(\cdot)$ are probability density and cumulative distribution functions of $c_i$, respectively.*

The above assumption holds for a wide class of distributions such as the ones with log-concave density functions (e.g., uniform).

**Theorem 2.** *Suppose Assumption 1 holds. For a given estimator $\hat{\theta} : \hat{\mathcal{X}}^n \times \mathbb{R}_+^n \times \mathbb{R}_+^n \to \mathbb{R}$, in the optimal two-part data acquisition mechanism, for a given vector of reported privacy sensitivities $\mathbf{c}$, the local and central privacy losses are the solution of*

$$\min_{\{\varepsilon^{(l)}\}_{i=1}^n, \{\varepsilon^{(c)}\}_{i=1}^n} \quad \gamma \text{MSE}(\varepsilon^{(l)}, \varepsilon^{(c)}, \hat{\theta}) + \sum_{i=1}^{n} \varepsilon_i^{(l)} \psi_i(c_i) + \sum_{i=1}^{n} \varepsilon_i^{(c)}(1 - \psi_i(c_i)). \tag{6}$$

*Proof Sketch of Theorem 2:* We introduce the following *interim functions*

$$t_i(c_i) = \mathbb{E}_{\mathbf{c}_{-i}}[t(c_i, \mathbf{c}_{-i})], \quad \varepsilon_i^{(l)}(c_i) = \mathbb{E}_{\mathbf{c}_{-i}}[\varepsilon_i^{(l)}(c_i, \mathbf{c}_{-i})], \quad \text{and } \varepsilon_i^{(c)}(c_i) = \mathbb{E}_{\mathbf{c}_{-i}}[\varepsilon_i^{(c)}(c_i, \mathbf{c}_{-i})].$$

We first establish a *payment identity* that determines the optimal payment in terms of the optimal local and central privacy losses. In particular, by evaluating the first order condition corresponding to the incentive compatibility constraint (4), we establish that this constraint holds if and only if

$$t_i(c_i) = t_i(0) + \varepsilon_i^{(c)}(c_i) - \varepsilon_i^{(c)}(0) + c_i(\varepsilon_i^{(l)}(c_i) - \varepsilon_i^{(c)}(c_i)) - \int_0^{c_i} (\varepsilon_i^{(l)}(z) - \varepsilon_i^{(c)}(z)) dz,$$

---

[2]Notice that changing the parameter $\gamma$ enables us to study a wide range of platform's objectives with differing relative weights between the estimation error and the total payments.

[3]Without this assumption, extending the results requires ironing technique of Myerson [1981].

and $\varepsilon_i^{(l)}(z) - \varepsilon_i^{(c)}(z)$ is weakly decreasing in $z$. We then plug in this payment identity back to the platform's objective, use the individual rationality constraint, and rewrite the platform's expected utility in terms of the privacy loss functions and the virtual cost of users. This is still a functional optimization problem in terms of $\varepsilon^{(l)}(\cdot)$ and $\varepsilon^{(c)}(\cdot)$. However, we establish that, under Assumption 1, we can solve this functional optimization point-wise (i.e., for any given $\mathbf{c}$). ∎

Theorem 2 highlights the tradeoff in the platform's problem: by decreasing the local privacy loss, the second term of the objective decreases (this term corresponds to the payment to users) while the first term (i.e., the mean squared error) increases. The role of the central privacy loss is more nuanced, and there are two cases. If the coefficient $1 - \psi_i(c_i)$ is non-negative, by decreasing the central privacy loss, the third term of the objective decreases while the first term increases. If the coefficient $1 - \psi_i(c_i)$ is negative, increasing the central privacy loss decreases both the third term and the first term. However, we cannot increase the central privacy loss level without limits because the central privacy loss level is always below the local privacy loss level. Therefore, the platform's optimal mechanism should find the "right" balance between these terms.

## 4    Optimal mechanism with two-part privacy guarantees for linear estimators

For the rest of the paper, we focus on linear estimators with Gaussian mechanism described in Section 2.1. The following is a direct corollary of Theorem 2.

**Corollary 1.** *Suppose Assumption 1 holds. For any reported vector of privacy sensitivities $\mathbf{c}$, the optimal local privacy loss levels are $\varepsilon_i^{(l)}(\mathbf{c}) = y_i^*$ and the optimal central privacy loss levels are*

$$\varepsilon_i^{(c)} = w_i^{*2} / \sum_{j=1}^n \frac{w_j^{*2}}{y_j^*} \text{ where } (w_1^*, \dots, w_n^*) \text{ and } (y_1^*, \dots, y_n^*) \text{ are the optimal solution of}$$

$$\min_{\mathbf{w}, \mathbf{y}} \text{VAR} \gamma \sum_{i=1}^n w_i^2 + \frac{\gamma \alpha}{2} \sum_{i=1}^n \frac{w_i^2}{y_i} + \sum_{i=1}^n (1 - \psi_i(c_i)) \frac{w_i^2}{\sum_{j=1}^n \frac{w_j^2}{y_j}} + \sum_{i=1}^n \psi_i(c_i) y_i \qquad (7)$$

$$\text{s.t. } w_i, y_i \geq 0, \text{ for all } i \in \mathcal{N} \text{ and } \sum_{i=1}^n w_i = 1.$$

Let us highlight the difference between our characterization and that of classic mechanism design (e.g., Myerson [1981]). In classic mechanism design, the designer's problem becomes linear optimization. However, in our setting, the designer's problem is a non-linear and non-convex optimization. This makes the problem of finding the optimal two-part data acquisition mechanism challenging. Before addressing this computational challenge, let us revisit the form of the Gaussian mechanism that we have adopted: the platform adds Gaussian noise locally and then outputs a convex combination of the privatized users' data without adding any noise centrally. More specifically, one may guess that the platform may benefit by having a central noise added to the final output in addition to the local noises. In the following subsection, we establish that there is another Gaussian mechanism for any Gaussian mechanism that only adds local noises and achieves a weakly lower cost.

### 4.1    Optimality of having only local noises in the Gaussian mechanism

The platform has the opportunity of adding Gaussian noise to both the raw data of each user and the final estimator and ex-ante one may guess that it is optimal to use both of these instruments. However, as we establish next, interestingly, in the optimal two-part data acquisition mechanism, it is always optimal to only add noises locally.

For a given vector of local privacy losses $(\varepsilon_1^{(l)}, \dots, \varepsilon_n^{(l)})$ and central privacy losses $(\varepsilon_1^{(c)}, \dots, \varepsilon_n^{(c)})$, a Gaussian mechanism with both local and central noises is of the form

$$\hat{\theta}(x_1, \dots, x_n) := \sum_{i=1}^n w_i \hat{x}_i + \mathcal{N}(0, \frac{\alpha}{2\varepsilon}) \text{ where } \sum_{i=1}^n w_i = 1 \text{ and } \hat{x}_i = x_i + \mathcal{N}(0, \frac{\alpha}{2\varepsilon_i^{(l)}}) \forall i \in \mathcal{N}.$$

**Proposition 1.** *In the optimal two-part data acquisition mechanism that adopts a Gaussian mechanism with both local and central noises, we have $\varepsilon = \infty$.*

**Algorithm 1:** Computing the optimal two-part private data acquisition mechanism

---

**Input:** The vector of privacy sensitivities $(c_1, \ldots, c_n)$

**for** $S \in \mathrm{Grid}\left([\underline{S}, \bar{S}], \delta\right)$ **do**

   Let

$$\nu_i = \frac{1}{\gamma \mathrm{VAR} + (1 - \psi_i(c_i))/S}, \; \zeta_i = \frac{\nu_i}{\sum_j \nu_j}, \; \xi_i = \zeta_i \left( \sum_{j=1}^n \nu_j (\sqrt{\psi_j(c_j)} - \sqrt{\psi_i(c_i)}) \right).$$

   Let $p = \left( \dfrac{\sum_{i=1}^n \zeta_i \sqrt{\psi_i(c_i)}}{S - \sum_{i=1}^n \sqrt{\psi_i(c_i)\xi_i}} \right)^2$.

   Let

$$w_i(S) = \frac{\nu_i + \nu_i \sum_j \nu_j \sqrt{\psi_j(c_j)p}}{\sum_j \nu_j} - \nu_i \sqrt{\psi_i(c_i)p}, \quad y_i(S) = w_i(S)\sqrt{\frac{p}{\psi_i(c_i)}},$$

   and $\mathrm{OBJ}(S)$ be the objective of Problem (7) evaluated for this solution.

**end**

**Output:** $\{y_i(S^*), w_i(S^*)\}_{i=1}^n$, where $(S^*) = \arg\min_{(S)} \mathrm{OBJ}(S)$.

---

Proposition 1 has an important implication in terms of the design of data market architecture when users have both central and local privacy costs: it is optimal to add noise locally! Adding a noise centrally to the final estimator has an advantage because the weights in the final estimator give the platform a lever to deliver heterogeneous central privacy guarantees to users. Despite this advantage, we establish that adding noise centrally is never optimal. This is because the platform prefers to add the noise locally to contribute to both central and local privacy guarantees delivered to users.

### 4.2 Computing the optimal privacy loss function

The implementation of the optimal two-part private data acquisition mechanism requires solving Problem (7), which is a non-convex program. However, we use the structure of the problem to develop a polynomial time algorithm to solve it approximately. To do so, we first replace $\sum_{i=1}^n w_i^2/y_i$ by an auxiliary variable $S$. Next, we consider the corresponding lagrangian problem. Using Karush-Kuhn-Tucker (KKT) conditions, we establish a number of relations between problems' parameters, $S$, and $p$, the lagrangian coefficient corresponding to $S = \sum_{i=1}^n w_i^2/y_i$. Furthermore, we develop upper and lower bounds for $S$. Finally, we do a grid search to find the approximate optimal solution.

**Theorem 3.** *For any vector of reported privacy sensitivities and $\epsilon > 0$, Algorithm 1 finds local privacy loss levels and the differentially private linear estimator of the two-part data acquisition mechanism whose cost (i.e., platform's objective) is at most $1 + \delta$ of the optimal cost in time $\mathrm{poly}(n, \frac{1}{\delta})$.*

Notice that the approximation factor in Theorem 3 depends on the underlying parameters. Therefore, we have a Polynomial Time Approximation Scheme (PTAS) for finding the optimal two-part data acquisition mechanism in the class of linear estimators.

In the Appendix, we provide a case study with two users to illustrate the performance of the optimal two-part data acquisition mechanism in terms of the guaranteed privacy levels and payments as functions of the reported privacy sensitivities.

## 5 Conclusion

In this paper we develop a unified framework to study the design of data acquisition mechanisms when users have both local and central privacy concerns and are heterogeneous in how they value these two privacy concerns. We use Rényi differential privacy to measure the privacy loss of users and first establish a minimax lower bound that motivates us to focus on linear estimators. We then establish a point-wise optimization problem whose solution fully characterizes the optimal data acquisition mechanism that constitute a payment scheme to compensate users for their privacy losses, a local privacy guarantee, and a central privacy guarantee all as a function of users' preferences for

local and central privacy concerns. We then focus on linear estimators, motivated by our optimality results, and establish that, even though the corresponding optimization problem is non-convex, the platform's problem admits a Polynomial Time Approximation Scheme. Finally, we focused on data acquisition to estimate mean population. However, our framework is more general and allows for considering other (potentially vector) estimates. In particular, our Theorem 2 converts the data acquisition mechanism design problem into a (potentially) non-convex optimization problem.

## 6 Acknowledgement

Alireza Fallah acknowledges support from the Apple Scholars in Ai/ML PhD fellowship.

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
