Figure 2: (a) the platform's objective and (b) the platform's estimation error as a function of $(c_1, c_2)$ for two users with $\gamma = 1$, $\alpha = 2$, and VAR $= 1/4$.

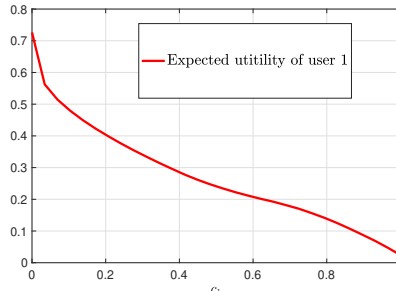

Figure 3: User 1's expected utility a function of $c_1$ for two users with $\gamma = 1$, $\alpha = 2$, and VAR $= 1/4$.

# A   Additional results and proofs

This appendix includes the additional results discussed in the text and the omitted proofs.

## A.1   An illustrative example

Here, we illustrate the results in a simple setting with two users. For $i = 1, 2$, we let $c_i$ be uniformly distributed over $[0, 1]$ so that the virtual value is $\psi_i(c_i) = 2c_i$. We also let $\gamma = 1$, $\alpha = 2$, and VAR $= 1/4$.

Figure 2 shows the platform's objective (i.e., the solution of Problem (7)) and its variance as a function of the privacy sensitivities. As we observe, higher privacy sensitivities (which means that users care more about the local privacy compared to central privacy) leads to a higher platform's cost and a higher estimation error. This is because guaranteeing local privacy is more demanding compared to central privacy. Figure 3 illustrates the expected utility of user 1 (similarly user 1) as a function of her privacy sensitivity. We observe that, unlike classical mechanism design settings, the utility is a continuous function of the user's type (as opposed to a threshold function). Again, we observe that higher privacy sensitivity implies better local privacy, which is more demanding and decreases the user's expected utility.

## A.2   Proofs

### Proof of Theorem 1

We establish the lower bound by using the Le Cam's method Yu [1997] which reduces the lower bound problem to a hypothesis testing problem between two distributions. To prove the lower bound, we need to show that for any $\hat{\theta} \in \mathcal{Q}((\varepsilon_i^{(l)})_{i=1}^n)$, we have

$$\sup_{P \in \mathcal{P}} \mathbb{E}_{(X_i \sim P)_{i=1}^n, \hat{\theta}} \left[ \left| \hat{\theta}((\mathcal{C}_i(X_i))_{i=1}^n) - \theta(P) \right|^2 \right] \geq c \min \left( \frac{1}{\sum_{i=1}^n \varepsilon_i^{(l)}}, 1 \right). \tag{8}$$

To show this result, we replace $\sup_{P \in \mathcal{P}}$ by an average over two carefully chosen distributions in $\mathcal{P}$. More formally, let $P_1$ and $P_2$ be two distributions of choice in $\mathcal{P}$ with $\gamma := \frac{1}{2}|\theta(P_1) - \theta(P_2)|$. Note that

$$\sup_{P \in \mathcal{P}} \mathbb{E}_{(X_i \sim P)_{i=1}^n, \hat{\theta}}\left[\left|\hat{\theta}((\mathcal{C}_i(X_i))_{i=1}^n) - \theta(P)\right|^2\right] \geq \frac{1}{2}\sum_{j=1}^2 \mathbb{E}_{(X_i \sim P_j)_{i=1}^n, \hat{\theta}}\left[\left|\hat{\theta}((\mathcal{C}_i(X_i))_{i=1}^n) - \theta(P_j)\right|^2\right]$$

$$= \frac{1}{2}\sum_{j=1}^2 \mathbb{E}_{Y \sim Q_j}\left[|Y - \theta(P_j)|^2\right], \tag{9}$$

where, for any $j \in \{1, 2\}$, $Q_j$ denotes the distribution of $\hat{\theta}((\mathcal{C}_i(X_i))_{i=1}^n)$ when $X_1, \cdots, X_n$ are drawn from $P_j$. We next lower bound the right-hand side of (9) by Markov's inequality

$$\sup_{P \in \mathcal{P}} \mathbb{E}_{(X_i \sim P)_{i=1}^n, \hat{\theta}}\left[\left|\hat{\theta}((\mathcal{C}_i(X_i))_{i=1}^n) - \theta(P)\right|^2\right] \geq \gamma^2 \frac{1}{2}\sum_{j=1}^2 \mathbb{P}\left(|Y - \theta(P_j)| \geq \gamma\right). \tag{10}$$

Now, consider a hypothesis testing problem with the goal of determining whether the underlying distribution is $P_1$ or $P_2$, given an observation of $Y$. One possible approach is to choose $j \in \{1, 2\}$ for which $|Y - \theta(P_j)|$ is smaller. It can be shown that the probability of an incorrect estimate by this approach is upper bounded by $\frac{1}{2}\sum_{j=1}^2 \mathbb{P}\left(|Y - \theta(P_j)| \geq \gamma\right)$ on the right-hand side of (10). Furthermore, a seminal result by Le Cam states that the infimum probability of incorrect decision among all possible mappings for the aforementioned hypothesis testing problem is given by $\frac{1}{2} - \frac{1}{2}\|Q_1 - Q_2\|_{\text{TV}}$. Therefore, we obtain the following lower bound

$$\mathcal{L}(\mathcal{P}, \mathcal{Q}, (\varepsilon_i^{(l)})_{i=1}^n) \geq \gamma^2 \left(\frac{1}{2} - \frac{1}{2}\|Q_1 - Q_2\|_{\text{TV}}\right). \tag{11}$$

Next, we provide an upper bound on $\|Q_1 - Q_2\|_{\text{TV}}$. To do so, we use the connection between the total variation distance and the Hellinger distance. Recall that the Hellinger distance between two distributions $\mu$ and $\nu$ is given by

$$d_{\text{hel}}(\mu, \nu)^2 := \int (\sqrt{d\mu(x)} - \sqrt{d\nu(x)})^2.$$

Hellinger distance has a number of well-known desirable properties. In particular, we use the following two:

- For any two distributions $\mu_1$ and $\mu_2$, we have
$$\|\mu - \nu\|_{\text{TV}} \leq d_{\text{hel}}(\mu, \nu). \tag{12}$$
- Let $\mu := \mu_1 \times \cdots \times \mu_n$ and $\nu := \nu_1 \times \cdots \times \nu_n$. Then,
$$d_{\text{hel}}(\mu, \nu)^2 = 2 - 2\prod_{i=1}^n (1 - \frac{1}{2}d_{\text{hel}}(\mu_i, \nu_i)^2). \tag{13}$$

Let us go back to the problem of upper bounding $\|Q_1 - Q_2\|_{\text{TV}}$. The following lemma is the key result in our proof:

**Lemma 2.** *Let $\alpha \geq 2$ and suppose $\mathcal{C}(.) : \mathcal{X} - \to \mathbb{R}$ is an $(\varepsilon, \alpha)$-RDP channel. For $j \in \{1, 2\}$, let $\nu_j$ be the distribution of $\mathcal{C}(X)$ when $X \sim \mu_j$. Then,*
$$d_{hel}(\nu_1, \nu_2)^2 \leq 2(e^\varepsilon - 1)\, d_{hel}(\mu_1, \mu_2)^2.$$

We defer the proof of Lemma 2 to the end of this section. Let us first complete the proof of lower bound using this lemma. Note that, by data processing inequality we have

$$\|Q_1 - Q_2\|_{\text{TV}} \leq \|(\mathcal{C}_i(X_i))_{i=1}^n - (\mathcal{C}_i(\tilde{X}_i))_{i=1}^n\|_{\text{TV}},$$

where $X_i \sim P_1$ and $\tilde{X}_i \sim P_2$. Next, using (12) and (13) implies

$$\|Q_1 - Q_2\|_{\text{TV}}^2 \leq \|(\mathcal{C}_i(X_i))_{i=1}^n - (\mathcal{C}_i(\tilde{X}_i))_{i=1}^n\|_{\text{TV}}^2$$

$$\leq d_{\text{hel}}((\mathcal{C}_i(X_i))_{i=1}^n, (\mathcal{C}_i(\tilde{X}_i))_{i=1}^n)^2$$

$$= 2 - 2\prod_{i=1}^n (1 - \frac{1}{2}d_{\text{hel}}(\mathcal{C}_i(X_i), \mathcal{C}_i(\tilde{X}_i))^2). \tag{14}$$

Next, note that, by Lemma 2, we have
$$d_{\text{hel}}(\mathcal{C}_i(X_i), \mathcal{C}_i(\tilde{X}_i))^2 \leq 2(e^{\varepsilon_i^{(l)}} - 1)\, d_{\text{hel}}(P_1, P_2)^2 \leq 4\varepsilon_i^{(l)}\, d_{\text{hel}}(P_1, P_2)^2,$$
where the last inequality follows from the fact that $\varepsilon_i^{(l)} \leq 1$. Plugging this back into (14), we obtain
$$\|Q_1 - Q_2\|_{\text{TV}}^2 \leq 2 - 2 \prod_{i=1}^n (1 - 2\varepsilon_i^{(l)}\, d_{\text{hel}}(P_1, P_2)^2). \tag{15}$$
Next, note that, for nonnegative $y_1, \cdots, y_n$ we have
$$\prod_{i=1}^n (1 - y_i) \geq 1 - \sum_{i=1}^n y_i.$$
To show this, we can first verify it for $n = 2$ and then it is straightforward to show it for any $n$ by induction. Using this inequality with $y_i = 2\varepsilon_i^{(l)}\, d_{\text{hel}}(P_1, P_2)^2$, we can further upper bound (15) by
$$\|Q_1 - Q_2\|_{\text{TV}}^2 \leq 4(\sum_{i=1}^n \varepsilon_i^{(l)})\, d_{\text{hel}}(P_1, P_2)^2. \tag{16}$$
Plugging this back into (11), we have
$$\mathcal{L}(\mathcal{P}, \mathcal{Q}, (\varepsilon_i^{(l)})_{i=1}^n) \geq \gamma^2 \left( \frac{1}{2} - \sqrt{\sum_{i=1}^n \varepsilon_i^{(l)}}\, d_{\text{hel}}(P_1, P_2) \right). \tag{17}$$
Next, we define $P_1$ and $P_2$ as
$$P_1(-1/2) = P_2(1/2) = \frac{1 + 2\gamma}{2}, \quad P_1(1/2) = P_2(-1/2) = \frac{1 - 2\gamma}{2}. \tag{18}$$
It is straightforward to verify that $|\theta(P_1) - \theta(P_2)| = 2\gamma$. Moreover, we have
$$d_{\text{hel}}(P_1, P_2)^2 = 2\left( \sqrt{\frac{1 + 2\gamma}{2}} - \sqrt{\frac{1 - 2\gamma}{2}} \right)^2 = 2(1 - \sqrt{1 - 4\gamma^2}) \leq 8\gamma^2,$$
where the last inequality follows from the fact that
$$1 - \sqrt{1 - 4\gamma^2} = \frac{4\gamma^2}{1 + \sqrt{1 - 4\gamma^2}} \leq 4\gamma^2.$$
Plugging this back into (17) implies
$$\mathcal{L}(\mathcal{P}, \mathcal{Q}, (\varepsilon_i^{(l)})_{i=1}^n) \geq \gamma^2 \left( \frac{1}{2} - \sqrt{8 \sum_{i=1}^n \varepsilon_i^{(l)}}\, \gamma \right). \tag{19}$$
Finally, setting
$$\gamma = \min \left( \frac{1}{4\sqrt{8 \sum_{i=1}^n \varepsilon_i^{(l)}}}, \frac{1}{2} \right)$$
completes the proof of lower bound.

To show the upper bound, first recall that a linear estimator with Gaussian mechanism is in the form of
$$\sum_{i=1}^n w_i \left( x_i + \mathcal{N}\left( 0, \frac{\alpha}{2\varepsilon_i^{(l)}} \right) \right). \tag{20}$$
The mean square error of this estimator is given by
$$\sum_{i=1}^n w_i^2 (\text{VAR} + \frac{\alpha}{2\varepsilon_i^{(l)}}) \leq \alpha \sum_{i=1}^n \frac{w_i^2}{\varepsilon_i^{(l)}},$$
where the last inequality uses the fact that $\alpha \geq 2$ and $\varepsilon_i^{(l)} \leq 1$. Finally, setting
$$w_i = \frac{\varepsilon_i^{(l)}}{\sum_{j=1}^n \varepsilon_j^{(l)}}$$
gives us the desired upper bound.

**Proof of Lemma 2**

Note that

$$d_{\text{hel}}(\nu_1, \nu_2)^2 = \int (\sqrt{\nu_1(z)} - \sqrt{\nu_2(z)})^2 dz = \int \frac{(\nu_1(z) - \nu_2(z))^2}{(\sqrt{\nu_1(z)} + \sqrt{\nu_2(z)})^2} dz$$

$$\leq \int \frac{(\nu_1(z) - \nu_2(z))^2}{\nu_1(z) + \nu_2(z)} dz. \tag{21}$$

Note that, for any $j \in \{1, 2\}$, we can cast $\nu_j(z)$ as

$$\nu_j(z) = \int_x \mathcal{C}(z|x) d\mu_j(x).$$

Moreover, for any $x' \in \mathcal{X}$, we have

$$\nu_1(z) - \nu_2(z) = \int_x \mathcal{C}(z|x)(d\mu_1(x) - d\mu_2(x)) = \int_x (\mathcal{C}(z|x) - \mathcal{C}(z|x'))(d\mu_1(x) - d\mu_2(x)).$$

Substituting these into (21), we have

$$d_{\text{hel}}(\nu_1, \nu_2)^2 \leq \int_z \frac{\left(\int_x (\mathcal{C}(z|x) - \mathcal{C}(z|x'))(d\mu_1(x) - d\mu_2(x))\right)^2}{\int_x \mathcal{C}(z|x)(d\mu_1(x) + d\mu_2(x))} dz. \tag{22}$$

Next, by Cauchy–Schwarz inequality, we obtain

$$\left(\int_x (\mathcal{C}(z|x) - \mathcal{C}(z|x'))(d\mu_1(x) - d\mu_2(x))\right)^2 \leq$$

$$\left(\int_x \mathcal{C}(z|x)(d\mu_1(x) + \mu_2(x))\right) \left(\int_x \frac{(\mathcal{C}(z|x) - \mathcal{C}(z|x'))^2}{\mathcal{C}(z|x)} \frac{(d\mu_1(x) - d\mu_2(x))^2}{d\mu_1(x) + d\mu_2(x)}\right).$$

Hence, using (22) and this inequality, we can further upper bound $d_{\text{hel}}(\nu_1, \nu_2)^2$ by

$$d_{\text{hel}}(\nu_1, \nu_2)^2 \leq \int_z \int_x \frac{(\mathcal{C}(z|x) - \mathcal{C}(z|x'))^2}{\mathcal{C}(z|x)} \frac{(d\mu_1(x) - d\mu_2(x))^2}{d\mu_1(x) + d\mu_2(x)} dz$$

$$= \int_x \left[\int_z \frac{(\mathcal{C}(z|x) - \mathcal{C}(z|x'))^2}{\mathcal{C}(z|x)} dz\right] \frac{(d\mu_1(x) - d\mu_2(x))^2}{d\mu_1(x) + d\mu_2(x)} \tag{23}$$

where the last equation follows from changing the order of integration using Fubini's theorem. Now, note that the first term on the right hand side of (23) can be cast as

$$\int_z \frac{(\mathcal{C}(z|x) - \mathcal{C}(z|x'))^2}{\mathcal{C}(z|x)} dz = \int_z \frac{\mathcal{C}(z|x')^2}{\mathcal{C}(z|x)} dz - 2 \int_z \mathcal{C}(z|x') dz + \int_z \mathcal{C}(z|x) dz$$

$$= \exp(D_2(\mathcal{C}(x')||\mathcal{C}(x))) - 1. \tag{24}$$

It is known that $D_\alpha(.||.)$ is nondecreasing in $\alpha$. Thus, using $\alpha \geq 2$, we obtain

$$\int_z \frac{(\mathcal{C}(z|x) - \mathcal{C}(z|x'))^2}{\mathcal{C}(z|x)} dz \leq e^\varepsilon - 1. \tag{25}$$

Plugging this back into (23), we have

$$d_{\text{hel}}(\nu_1, \nu_2)^2 \leq (e^\varepsilon - 1) \int_x \frac{(d\mu_1(x) - d\mu_2(x))^2}{d\mu_1(x) + d\mu_2(x)}. \tag{26}$$

To complete the proof, we just need to show that

$$\int_x \frac{(d\mu_1(x) - d\mu_2(x))^2}{d\mu_1(x) + d\mu_2(x)} \leq 2d_{\text{hel}}(\mu_1, \mu_2)^2.$$

To do so, note that

$$\int_x \frac{(d\mu_1(x) - d\mu_2(x))^2}{d\mu_1(x) + d\mu_2(x)} \leq 2 \int_x \frac{(d\mu_1(x) - d\mu_2(x))^2}{(\sqrt{d\mu_1(x)} + \sqrt{d\mu_2(x)})^2}$$

$$= 2 \int_x \left(\sqrt{d\mu_1(x)} - \sqrt{d\mu_2(x)}\right)^2 = 2d_{\text{hel}}(\mu_1, \mu_2)^2.$$

This concludes the proof of lemma 2 and hence the proof of Theorem 1. ∎

**Proof of Theorem 2**

Recall the iterim quantities

$$t_i(c_i) = \mathbb{E}_{\mathbf{c}_{-i}}\left[t(c_i, \mathbf{c}_{-i})\right],$$

$$\varepsilon_i^{(l)}(c_i) = \mathbb{E}_{\mathbf{c}_{-i}}\left[\varepsilon_i^{(l)}(c_i, \mathbf{c}_{-i})\right], \text{ and}$$

$$\varepsilon_i^{(c)}(c_i) = \mathbb{E}_{\mathbf{c}_{-i}}\left[\varepsilon_i^{(c)}(c_i, \mathbf{c}_{-i})\right] \text{ for all } i \in \mathcal{N}, c_i.$$

Using these quantities, the incentive compatibility constraint becomes

$$t_i(c_i) - c_i\varepsilon_i^{(l)}(c_i) - (1-c_i)\varepsilon_i^{(c)}(c_i) \geq t_i(c_i') - c_i\varepsilon_i^{(l)}(c_i') - (1-c_i)\varepsilon_i^{(c)}(c_i').$$

By equating the derivative of the right-hand side with respect to $c_i'$ at $c_i$ to zero, we obtain

$$t_i'(c_i) - c_i\left(\varepsilon_i'^{(l)}(c_i) - \varepsilon_i'^{(c)}(c_i)\right) - \varepsilon_i'^{(c)}(c_i) = 0.$$

This equation gives us the derivative of the payment in terms of the privacy loss levels. By taking the integral of this expression we obtain

$$t_i(c_i) = t_i(0) + \int_0^{c_i}\left(\varepsilon_i'^{(c)}(z) + z\left(\varepsilon_i'^{(l)}(z) - \varepsilon_i'^{(c)}(z)\right)\right)dz$$

$$= t_i(0) + \varepsilon_i^{(c)}(c_i) - \varepsilon_i^{(c)}(0) + c_i\left(\varepsilon_i^{(l)}(c_i) - \varepsilon_i^{(c)}(c_i)\right) - \int_0^{c_i}\left(\varepsilon_i^{(l)}(z) - \varepsilon_i^{(c)}(z)\right)dz. \quad (27)$$

We next show that the payment in (27) together with a weakly decreasing $\varepsilon_i^{(l)}(z) - \varepsilon_i^{(c)}(z)$ guarantees that the incentive compatibility constraint. To see this, we consider two possibilities depending on whether $c_i'$ is larger or smaller than $c_i$:

- For $c_i' \geq c_i$, by using the payment in (27), the incentive compatibility constraint becomes equivalent to

$$\left(\varepsilon_i^{(l)}(c_i') - \varepsilon_i^{(c)}(c_i')\right)(c_i - c_i') \geq \int_{c_i'}^{c_i}\left(\varepsilon_i^{(l)}(z) - \varepsilon_i^{(c)}(z)\right)dz,$$

  which holds because $\varepsilon_i^{(l)}(z) - \varepsilon_i^{(c)}(z)$ is weakly decreasing in $z$.

- For $c_i' \leq c_i$, by using the payment in (27), the incentive compatibility constraint becomes equivalent to

$$\left(\varepsilon_i^{(l)}(c_i') - \varepsilon_i^{(c)}(c_i')\right)(c_i - c_i') \leq \int_{c_i}^{c_i'}\left(\varepsilon_i^{(l)}(z) - \varepsilon_i^{(c)}(z)\right)dz,$$

  which, again, holds because $\varepsilon_i^{(l)}(z) - \varepsilon_i^{(c)}(z)$ is weakly decreasing in $z$. This completes one direction of the proof.

To see the other direction, notice that using the first order condition for the incentive compatibility constraints, imply (27). To see the monotonicity, notice that the incentive compatibility implies

$$\varepsilon_i^{(c)}(c_i) + c_i\left(\varepsilon_i^{(l)}(c_i) - \varepsilon_i^{(c)}(c_i)\right) - t_i(c_i) \leq \varepsilon_i^{(c)}(c_i') + c_i\left(\varepsilon_i^{(l)}(c_i') - \varepsilon_i^{(c)}(c_i')\right) - t_i(c_i').$$

and

$$\varepsilon_i^{(c)}(c_i') + c_i'\left(\varepsilon_i^{(l)}(c_i') - \varepsilon_i^{(c)}(c_i')\right) - t_i(c_i') \leq \varepsilon_i^{(c)}(c_i) + c_i'\left(\varepsilon_i^{(l)}(c_i) - \varepsilon_i^{(c)}(c_i)\right) - t_i(c_i).$$

The summation of these two inequalities yields

$$\left(\left(\varepsilon_i^{(l)}(c_i) - \varepsilon_i^{(c)}(c_i)\right) - \left(\varepsilon_i^{(l)}(c_i') - \varepsilon_i^{(c)}(c_i')\right)\right)(c_i - c_i') \leq 0,$$

that proves $\varepsilon_i^{(l)}(\cdot) - \varepsilon_i^{(c)}(\cdot)$ is weakly decreasing.

We next evaluate the individual rationality constraint. Using (27), we can rewrite this constraint as

$$t_i(0) \geq \varepsilon_i^{(c)}(0) + \int_0^{c_i} \left( \varepsilon_i^{(l)}(z) - \varepsilon_i^{(c)}(z) \right) dz \quad \text{for all } c_i. \tag{28}$$

Using $\varepsilon_i^{(l)}(z) \geq \varepsilon_i^{(c)}(z)$ for all $z$, this inequality means that it only needs to hold for $c_i = \infty$. Hence, we could cast $t_i(0)$ as

$$t_i(0) = \varepsilon_i^{(c)}(0) + \int_0^\infty \left( \varepsilon_i^{(l)}(z) - \varepsilon_i^{(l)}(z) \right) dz.$$

Plugging this back in (27) yields

$$t_i(c_i) = \varepsilon_i^{(c)}(c_i) + c_i \left( \varepsilon_i^{(l)}(c_i) - \varepsilon_i^{(c)}(c_i) \right) + \int_{c_i} \left( \varepsilon_i^{(l)}(z) - \varepsilon_i^{(c)}(z) \right) dz.$$

which is the optimal payment when $\varepsilon^{(l)}(\cdot)$ is decreasing.

With this optimal payment, the expected payment becomes

$$\mathbb{E}_{c_i}\left[ t_i(c_i) \right] = \mathbb{E}_{c_i}[\varepsilon_i^{(c)}(c_i)] + \mathbb{E}_{c_i}\left[ c_i \left( \varepsilon_i^{(l)}(c_i) - \varepsilon_i^{(c)}(c_i) \right) + \int_{z=c_i} \left( \varepsilon_i^{(l)}(z) - \varepsilon_i^{(c)}(z) \right) dz \right]$$

$$= \mathbb{E}_{c_i}[\varepsilon_i^{(c)}(c_i)]$$

$$+ \int_{\mathbf{z}_{-i}} \int_{z_i} \left( z_i \left( \varepsilon_i^{(l)}(z_i, \mathbf{z}_{-i}) - \varepsilon_i^{(c)}(z_i, \mathbf{z}_{-i}) \right) + \int_{y_i=z_i} \left( \varepsilon_i^{(l)}(y, \mathbf{z}_{-i}) - \varepsilon_i^{(c)}(y, \mathbf{z}_{-i}) \right) dy_i \right) f_i(z_i) dz_i f_{-i}(\mathbf{z}_{-i}) d\mathbf{z}_{-i}$$

$$\overset{(a)}{=} \mathbb{E}_{c_i}[\varepsilon_i^{(c)}(c_i)]$$

$$+ \int_{\mathbf{z}_{-i}} \int_{z_i} \left( z_i \left( \varepsilon_i^{(l)}(z_i, \mathbf{z}_{-i}) - \varepsilon_i^{(c)}(z_i, \mathbf{z}_{-i}) \right) + \left( \varepsilon_i^{(l)}(z_i, \mathbf{z}_{-i}) - \varepsilon_i^{(c)}(z_i, \mathbf{z}_{-i}) \right) \frac{F_i(z_i)}{f_i(z_i)} \right) f_i(z_i) dz_i f_{-i}(\mathbf{z}_{-i}) d\mathbf{z}_{-i}$$

$$= \mathbb{E}_{c_i}[\varepsilon_i^{(c)}(c_i)] + \int_{\mathbf{z}} \left( z_i + \frac{F_i(z_i)}{f_i(z_i)} \right) \left( \varepsilon_i^{(l)}(\mathbf{z}) - \varepsilon_i^{(c)}(\mathbf{z}) \right) f(\mathbf{z}) d\mathbf{z}, \tag{29}$$

where (a) follows from changing the order of the integrals. Substituting equation (29) in the platform's objective function yields

$$\mathbb{E}_{\mathbf{c}}\left[ \gamma \mathrm{MSE}(\boldsymbol{\varepsilon}^{(l)}(\mathbf{c}), \boldsymbol{\varepsilon}^{(c)}(\mathbf{c}), \hat{\theta}) + \sum_{i=1}^n t_i(\mathbf{c}) \right]$$

$$= \mathbb{E}_{\mathbf{c}}\left[ \gamma \mathrm{MSE}(\boldsymbol{\varepsilon}^{(l)}(\mathbf{c}), \boldsymbol{\varepsilon}^{(c)}(\mathbf{c}), \hat{\theta}) + \sum_{i=1}^n \psi_i(c_i) \varepsilon_i^{(l)}(\mathbf{c}) + \sum_{i=1}^n \left( 1 - \psi_i(c_i) \right) \varepsilon_i^{(c)}(\mathbf{c}) \right].$$

Notice that the maximizer of the above objective is the optimal local and central privacy losses, provided that $\varepsilon_i^{(l)}(\cdot) - \varepsilon_i^{(c)}(\cdot)$ is decreasing. For a given privacy sensitivity vector $\mathbf{c}$, let us consider the point-wise minimization given by

$$\min_{\{\boldsymbol{\varepsilon}^{(l)}\}_{i=1}^n, \{\boldsymbol{\varepsilon}^{(c)}\}_{i=1}^n} \gamma \mathrm{MSE}(\boldsymbol{\varepsilon}^{(l)}, \boldsymbol{\varepsilon}^{(c)}, \hat{\theta}) + \sum_{i=1}^n \varepsilon_i^{(l)} \psi_i(c_i) + \sum_{i=1}^n \varepsilon_i^{(c)} \left( 1 - \psi_i(c_i) \right). \tag{30}$$

This point-wise optimization clearly finds the optimal $\varepsilon^{(l)}(\cdot)$ and $\varepsilon^{(c)}(\cdot)$, but the issue is that the corresponding $\varepsilon_i^{(l)}(\cdot) - \varepsilon_i^{(c)}(\cdot)$ may not be decreasing. We next show that under Assumption 1 this is always the case.

Let $\{\varepsilon^{(l)}\}_{i=1}^n$ and $\{\varepsilon^{(c)}\}_{i=1}^n$ be the solution of optimization problem (30) for $c_1, \ldots, c_n$. Now, suppose we increase one of the $c_i$'s, which, without loss of generality, we assume is the first one. Let $c_1' > c_1$ and $c_i' = c_i$ for $i = 2, \ldots, n$ and suppose $\{\varepsilon'^{(l)}\}_{i=1}^n, \{\varepsilon'^{(c)}\}_{i=1}^n$ is the corresponding optimal solution of optimization problem (30). The optimality condition implies that

$$\gamma \mathrm{MSE}(\boldsymbol{\varepsilon}^{(l)}, \boldsymbol{\varepsilon}^{(c)}, \hat{\theta}) + \sum_{i=1}^n \varepsilon_i^{(l)} \psi_i(c_i) + \sum_{i=1}^n \varepsilon_i^{(c)} \left( 1 - \psi_i(c_i) \right)$$

$$\leq \gamma \mathrm{MSE}(\boldsymbol{\varepsilon}'^{(l)}, \boldsymbol{\varepsilon}'^{(c)}, \hat{\theta}) + \sum_{i=1}^n \varepsilon_i'^{(l)} \psi_i(c_i) + \sum_{i=1}^n \varepsilon_i'^{(c)} \left( 1 - \psi_i(c_i) \right)$$

and

$$\gamma\mathrm{MSE}(\boldsymbol{\varepsilon}'^{(l)}, \boldsymbol{\varepsilon}'^{(c)}, \hat{\theta}) + \sum_{i=1}^{n} {\varepsilon'}_i^{(l)}\psi_i(c_i') + \sum_{i=1}^{n} {\varepsilon'}_i^{(c)}\left(1 - \psi_i(c_i)\right)$$

$$\leq \gamma\mathrm{MSE}(\boldsymbol{\varepsilon}^{(l)}, \boldsymbol{\varepsilon}^{(c)}, \hat{\theta}) + \sum_{i=1}^{n} \varepsilon_i^{(l)}\psi_i(c_i') + \sum_{i=1}^{n} \varepsilon_i^{(c)}\left(1 - \psi_i(c_i)\right)$$

The summation of both sides of these inequalities, together with $c_i = c_i'$ for $i = 2, \ldots, n$, results in

$$\left(\left(\varepsilon_1^{(l)} - \varepsilon_1^{(c)}\right) - \left({\varepsilon'}_1^{(l)} - {\varepsilon'}_1^{(c)}\right)\right)(\psi_1(c_1) - \psi_1(c_1')) \leq 0.$$

Assumption 1 and the above inequality establishes that the solution of problem (30) is weakly decreasing in the privacy sensitivity. ∎

**Proof of Corollary 1**

The proof follows by invoking Theorem 2 and noting that with

$$\hat{\theta} = \sum_{i=1}^{n} w_i\left(x_i + \mathcal{N}\left(0, \frac{\alpha}{2\varepsilon_i^{(l)}}\right)\right)$$

we have

$$\varepsilon_i^{(c)} = \frac{w_i^2}{\sum_{j=1}^{n} \frac{w_j^2}{\varepsilon^{(l)2}_j}}$$

and

$$\mathrm{MSE}(\boldsymbol{\varepsilon}^{(l)}, \boldsymbol{\varepsilon}^{(c)}, \hat{\theta}) = \mathrm{VAR}\sum_{i=1}^{n} w_i^2 + \sum_{i=1}^{n} w_i^2 \frac{\alpha}{2\varepsilon_i^{(l)}}.$$

This completes the proof. ∎

**Proof of Proposition 1**

With a Gaussian mechanism that adopts both local and central noises, using a similar argument to that of Theorem 2 and Corollary 1, the optimal central privacy loss levels are

$$\varepsilon_i^{(c)} = \frac{w_i^{*2}}{\sum_{j=1}^{n} \frac{w_j^{*2}}{y_j^*} + \frac{1}{\varepsilon}},$$

where $(w_1^*, \ldots, w_n^*)$, $(y_1^*, \ldots, y_n^*)$, and $\varepsilon$ are the optimal solution of

$$\min_{\mathbf{w}, \mathbf{y}, \varepsilon} \ \mathrm{VAR}\gamma\sum_{i=1}^{n} w_i^2 + \frac{\gamma\alpha}{2}\sum_{i=1}^{n} \frac{w_i^2}{y_i} + \frac{\alpha\gamma}{2\varepsilon} + \sum_{i=1}^{n}(1 - \psi_i(c_i))\frac{w_i^2}{\sum_{j=1}^{n}\frac{w_j^2}{y_j} + \frac{1}{\varepsilon}} + \sum_{i=1}^{n}\psi_i(c_i)y_i$$

$$\text{s.t. } w_i, y_i \geq 0, \text{ for all } i \in \mathcal{N}$$

$$\sum_{i=1}^{n} w_i = 1.$$

For any solution of the above optimization problem we define the following alternative solution:

$$y_i' = \frac{w_i^2}{\frac{w_i^2}{y_i} + \frac{w_i}{\varepsilon}}, \varepsilon' = \infty, \text{ and } w_i' = w_i \quad \text{for all } i \in \mathcal{N}.$$

We have

$$\mathrm{VAR}\gamma \sum_{i=1}^{n} w_i'^2 + \frac{\gamma\alpha}{2} \sum_{i=1}^{n} \frac{w_i'^2}{y_i'} + \sum_{i=1}^{n} (1 - \psi_i(c_i)) \frac{w_i'^2}{\sum_{j=1}^{n} \frac{w_j'^2}{y_j'}} + \sum_{i=1}^{n} \psi_i(c_i) y_i'$$

$$\overset{(a)}{=} \mathrm{VAR}\gamma \sum_{i=1}^{n} w_i^2 + \frac{\gamma\alpha}{2} \sum_{i=1}^{n} \frac{w_i^2}{y_i} + \frac{\alpha\gamma}{2\varepsilon} + \sum_{i=1}^{n} (1 - \psi_i(c_i)) \frac{w_i^2}{\sum_{j=1}^{n} \frac{w_j^2}{y_j} + \frac{1}{\varepsilon}} + \sum_{i=1}^{n} \psi_i(c_i) y_i'$$

$$\overset{(b)}{\leq} \mathrm{VAR}\gamma \sum_{i=1}^{n} w_i^2 + \frac{\gamma\alpha}{2} \sum_{i=1}^{n} \frac{w_i^2}{y_i} + \frac{\alpha\gamma}{2\varepsilon} + \sum_{i=1}^{n} (1 - \psi_i(c_i)) \frac{w_i^2}{\sum_{j=1}^{n} \frac{w_j^2}{y_j} + \frac{1}{\varepsilon}} + \sum_{i=1}^{n} \psi_i(c_i) y_i,$$

where (a) follows the construction of the new solution and (b) follows from

$$y_i' = \frac{w_i^2}{\frac{w_i^2}{y_i} + \frac{w_i}{\varepsilon}} \leq y_i \quad \text{for all } i \in \mathcal{N}.$$

This completes the proof.

**Proof of Theorem 3**

Consider the optimization problem

$$\min_{\mathbf{w},\mathbf{y}} \ \mathrm{VAR}\gamma \sum_{i=1}^{n} w_i^2 + \frac{\gamma\alpha}{2} \sum_{i=1}^{n} \frac{w_i^2}{y_i} + \sum_{i=1}^{n} \frac{w_i^2}{\sum_{j=1}^{n} \frac{w_j^2}{y_j}} (1 - \psi_i(c_i)) + \sum_{i=1}^{n} \psi_i(c_i) y_i \tag{31}$$

$$\text{s.t. } w_i, y_i \geq 0, \text{ for all } i \in \mathcal{N}$$

$$\sum_{i=1}^{n} w_i = 1.$$

We can rewrite this optimization problem as

$$\min_{\mathbf{w},\mathbf{y},S} \ \mathrm{VAR}\gamma \sum_{i=1}^{n} w_i^2 + \frac{\gamma\alpha}{2} S + \frac{1}{S} \sum_{i=1}^{n} w_i^2 (1 - \psi_i(c_i)) + \sum_{i=1}^{n} \psi_i(c_i) y_i$$

$$\text{s.t. } w_i, y_i \geq 0, \text{ for all } i \in \mathcal{N}$$

$$\sum_{i=1}^{n} w_i = 1,$$

$$\sum_{i=1}^{n} \frac{w_i^2}{y_i} = S.$$

Let us fix $S$. The Lagrangian of this optimization problem becomes

$$\sum_{i=1}^{n} w_i^2 \left( \mathrm{VAR}\gamma + \frac{1 - \psi_i(c_i)}{S} \right) + \sum_{i=1}^{n} \psi_i(c_i) y_i + p \left( \sum_{i=1}^{n} \frac{w_i^2}{y_i} - S \right) - q \left( \sum_{i=1}^{n} w_i - 1 \right)$$

$$- \sum_{i=1}^{n} u_i w_i - \sum_{i=1}^{n} v_i y_i,$$

where $u_i, v_i \geq 0$ and $u_i w_i = v_i y_i = 0$, for all $i$.

Equating the derivative with respect to $w_i$ to zero, yields

$$2 w_i \left( \mathrm{VAR}\gamma + \frac{1 - \psi_i(c_i)}{S} \right) + \frac{2 p w_i}{y_i} - q = u_i. \tag{32}$$

Hence, if $w_i^* > 0$, then $u_i = 0$ which implies

$$2 \left( \mathrm{VAR}\gamma + \frac{1 - \psi_i(c_i)}{S} + \frac{p}{y_i} \right) w_i = q. \tag{33}$$

On the other hand, if $w_i^* = 0$, then $u_i = -q \geq 0$.

Equating the derivative with respect to $y_i$ to zero implies

$$\psi_i(c_i) - \frac{p w_i^2}{y_i^2} - v_i.$$

Hence, if $y_i^* = 0$, then $w_i^* = 0$ and $v_i = \psi_i(c_i)$. On ther other hand, if $y_i > 0$, then $v_i = 0$ and we have

$$w_i = \sqrt{\frac{\psi_i(c_i)}{p}} y_i. \tag{34}$$

Now, we claim there is no $i$ for which $w_i^* = 0$ (and hence there is no $i$ for which $y_i^* = 0$). Assume this is not the case, and hence there exists some $i_0$ for which $w_{i_0}^* = 0$. Therefore, as we established earlier, we have $q = -u_{i_0} \leq 0$. On the other hand, note that there exists $j$ for which $w_j^* > 0$. Hence, for that $y_j^* > 0$ as well. Therefore, using (33), along with the fact that $q \leq 0$, we should have

$$\frac{1 - \psi_j(c_j)}{S} + \frac{p}{y_j^*} \leq 0,$$

which implies

$$\frac{\psi_j(c_j) - 1}{S} \geq \frac{p}{y_j^*}.$$

Hence, using $S \geq (w_j^*)^2 / y_j^*$, we have

$$\psi_j(c_j) - 1 \geq S \frac{p}{y_j^*} \geq \frac{(w_j^*)^2 p}{y_j^*}.$$

However, since $y_j^* > 0$, by (34), the right hand side is equal to $\psi_j(c_j)$, which implies $\psi_j(c_j) - 1 \geq \psi_j(c_j)$ which is a contradiction! As a result, (33) and (34) hold for all $i$.

By invoking (34) in (33), we obtain

$$w_i = \frac{1}{\mathrm{VAR}\gamma + \frac{1 - \psi_i(c_i)}{S}} \left( \frac{q}{2} - \sqrt{\psi_i(c_i)p} \right).$$

To simplify the analysis, we define the interim variable

$$\nu_i = \frac{1}{\gamma \mathrm{VAR} + (1 - \psi_i(c_i))/S}. \tag{35}$$

Taking summation of the above equation for $i = 1, \ldots, n$ and using $\sum_{i=1}^n w_i = 1$, we obtain

$$\frac{q}{2} = \frac{1 + \sum_j \nu_j \sqrt{\psi_j(c_j)p}}{\sum_j \nu_j},$$

which together with (32) results in

$$w_i = \frac{\nu_i}{\sum_j \nu_j} + \frac{\nu_i}{\sum_j \nu_j} \left( \sum_{j=1}^n \nu_j (\sqrt{\psi_j(c_j)p} - \sqrt{\psi_i(c_i)p}) \right). \tag{36}$$

Therefore, by using (34) and (36), once we have $S$ and $p$, we can find $y_i$ and $w_i$ for all $i \in \mathcal{N}$.

Next, we derive a relation between $S$ and $p$. Note that (36) implies that $w_i$ can be cast as

$$\zeta_i(S) + \xi_i(S)\sqrt{p}$$

with

$$\zeta_i(S) = \frac{\nu_i}{\sum_j \nu_j} \quad \text{and} \quad \xi_i(S) = \frac{\nu_i}{\sum_j \nu_j} \left( \sum_{j=1}^n \nu_j (\sqrt{\psi_j(c_j)} - \sqrt{\psi_i(c_i)}) \right).$$

Using (34), we have

$$S = \sum_{i=1}^{n} \frac{w_i^2}{y_i} = \sum_{i=1}^{n} w_i \frac{\sqrt{\psi_i(c_i)}}{\sqrt{p}} = \sum_{i=1}^{n} \frac{\zeta_i(S)\sqrt{\psi_i(c_i)}}{\sqrt{p}} + \sum_{i=1}^{n} \sqrt{\psi_i(c_i)}\xi_i(S).$$

This implies

$$p = \left( \frac{\sum_{i=1}^{n} \zeta_i(S)\sqrt{\psi_i(c_i)}}{S - \sum_{i=1}^{n} \sqrt{\psi_i(c_i)}\xi_i(S)} \right)^2. \tag{37}$$

We next show that we can search over a grid to find the approximately optimal $S$. In this regard, we derive a lower and upper bound on the optimal $S$.

To do so, first note that the objective function (31) is given by

$$\begin{aligned} \text{OBJ} &= \text{VAR}\gamma \sum_{i=1}^{n} w_i^2 + \frac{\gamma\alpha}{2}S + \frac{1}{S}\sum_{i=1}^{n} w_i^2\left(1 - \psi_i(c_i)\right) + \sum_{i=1}^{n} \psi_i(c_i)y_i \\ &\geq \frac{\gamma\alpha}{2}S + \frac{1}{S}\sum_{i=1}^{n} w_i^2 + \sum_{i=1}^{n} \psi_i(c_i)(y_i - \frac{w_i^2}{S}). \end{aligned}$$

It is straightforward to see $y_i \geq \frac{w_i^2}{S}$ for all $i$, and thus, we have

$$\text{OBJ} \geq \frac{\gamma\alpha}{2}S + \frac{1}{S}\sum_{i=1}^{n} w_i^2. \tag{38}$$

Using (38) along with the fact that Cauchy–Schwarz inequality implies $\sum_{i=1}^{n} w_i^2 \geq 1/n$, we have

$$\text{Optimal objective (OPT)} \geq \frac{\gamma\alpha}{2}S^* + \frac{1}{nS^*}.$$

As a result, we have

$$\frac{\text{OPT}}{\gamma\alpha/2} \geq S^* \geq \frac{1}{\text{OPT}n}. \tag{39}$$

Letting $y_i = 1$, $w_i = \frac{1}{n}$, and $\varepsilon = 1$ in the objective of Problem (31) gives us an upper bound on the optimal objective OPT. Let us denote this upper bound by $M$. We have

$$\frac{M}{\gamma\alpha/2} \geq S^* \geq \frac{1}{Mn}. \tag{40}$$

Therefore, we obtain an approximate optimal solution by grid search. This provides an $O(\delta)$ optimal solution for the platform's problem because the objective of Problem (31) is Lipschitz continuous. ∎

### A.3 Revelation principle

Consider the strategy of user $i$ is a function of its relative privacy sensitivity shown by $\beta_i(c_i)$. For a given estimator $\hat{\theta}$ and mechanism $(\varepsilon^{(l)}, \varepsilon^{(c)}, \mathbf{t})$, the action profile $\{\beta_i(\cdot)\}_{i=1}^{n}$ is an equilibrium if

$$\begin{aligned} &\mathbb{E}_{\mathbf{c}_{-i}}\left[ t_i(\beta_{-\mathbf{i}}(\mathbf{c}_{-\mathbf{i}}), \beta_\mathbf{i}(\mathbf{c_i})) - \mathbf{c_i}\varepsilon_\mathbf{i}^{(\mathbf{l})}(\beta_{-\mathbf{i}}(\mathbf{c}_{-\mathbf{i}}), \beta_\mathbf{i}(\mathbf{c_i})) - \varepsilon_\mathbf{i}^{(\mathbf{c})}(\beta_{-\mathbf{i}}(\mathbf{c}_{-\mathbf{i}}), \beta_\mathbf{i}(\mathbf{c_i})) \right] \\ &\geq \mathbb{E}_{\mathbf{c}_{-i}}\left[ t_i(\beta_{-\mathbf{i}}(\mathbf{c}_{-\mathbf{i}}), \beta_\mathbf{i}'(\mathbf{c_i})) - \mathbf{c_i}\varepsilon_\mathbf{i}^{(\mathbf{l})}(\beta_{-\mathbf{i}}(\mathbf{c}_{-\mathbf{i}}), \beta_\mathbf{i}'(\mathbf{c_i})) - \varepsilon_\mathbf{i}^{(\mathbf{c})}(\beta_{-\mathbf{i}}(\mathbf{c}_{-\mathbf{i}}), \beta_\mathbf{i}'(\mathbf{c_i})) \right] \end{aligned}$$

for all $i \in \mathcal{N}, c_i, \beta_i'(\cdot)$. By letting $(\tilde{\varepsilon}^{(l)}, \tilde{\varepsilon}^{(c)}, \tilde{\mathbf{t}})$ be such that $\tilde{\varepsilon}_i^{(l)}(c_1, \ldots, c_n) = \varepsilon_i^{(l)}(\beta_1(c_1), \ldots, \beta_n(c_n))$, $\tilde{\varepsilon}_i^{(c)}(c_1, \ldots, c_n) = \varepsilon_i^{(c)}(\beta_1(c_1), \ldots, \beta_n(c_n))$, and $\tilde{t}_i(c_1, \ldots, c_n) = t_i(\beta_1(c_1), \ldots, \beta_n(c_n))$, the users will report truthfully and that the platform's objective is the same as the original mechanism. This establishes the revelation principle. ∎