# OpenReview forum: "Bridging Central and Local Differential Privacy in Data Acquisition Mechanisms"
_NeurIPS.cc/2022/Conference — NeurIPS 2022 Accept_

### Official Review · Reviewer_BCdV · 2022-07-11

**Rating:** 5
**Confidence:** 3
**Soundness:** 3 good
**Presentation:** 3 good
**Contribution:** 3 good

**Summary:**

The authors study the design of optimal Bayesian data acquisition mechanisms for a platform interested in estimating the mean of distribution by collecting data from privacy-conscious users. The authors assume users have heterogeneous sensitivities for local and central differential privacy measures and the users share their data in exchange for a payment that compensates for their privacy losses. The authors establish minimax lower bounds for the estimation error and show that a linear estimator is (near) optimal. A Bayesian setting is considered to turn the design of an optimal data acquisition mechanism into a nonconvex optimization problem.

**Questions:**

1. From my point of view, the assumption that the distribution of users’ sensitivities is known may be a strict condition. Can the authors give some examples to show that the assumption may be satisfied in some cases?

2. I am a little confused about the privacy sensitivities (c_1, …, c_n). The authors mentioned that it should be unknown to the platform, however, in line 256, it is assumed that “the privacy sensitivity is c_i and she reports c’_i”, also, in algorithm 1, the privacy sensitivities are also assumed to be given. Have the sensitivities been reported by the users? If not, how to estimate the privacy sensitivities when optimization?

3. In line 330, a Gaussian mechanism is given. What is the sensitivity L_i for the linear estimation of the sample mean?


**Limitations:**

The authors have adequately addressed the limitations.

**Strengths And Weaknesses:**

The manuscript is well organized and includes enough information needed to support the claims it makes. The proposed method is novel in the sense that the authors consider the heterogeneous sensitivities of users, and use a Bayesian setting to cast the mechanism designing problem as an optimization problem.

---

> ### Author Response · Authors · 2022-08-02
> **Response to Reviewer BCdV**
>
> *Thank you very much for your time in reviewing our paper and for providing valuable feedback. We are grateful for your positive assessment. Below, we respond to your comments in the order they appear in your report. We are grateful for giving us the opportunity to answer your questions, and we hope that the following addresses your questions.*
>
> **From my point of view, the assumption that the distribution of users’ sensitivities is known may be a strict condition. Can the authors give some examples to show that the assumption may be satisfied in some cases?**
>
> *As you mentioned, we assume that the distributions of the underlying privacy sensitivities are publicly known, while the realized privacy sensitivity of a user is private, and, because of this, we design an incentive compatible mechanism to elicit the true privacy sensitivity. The assumption that the underlying distribution of types is known is widely adopted in Bayesian mechanism design (as opposed to prior-free and worst-case mechanism design). Please see Myerson (1981) and, more broadly, Harsanyi (1967–1968) and Morris (94) for foundations of having known underlying priors. In practice, for instance, consider a movie recommendation platform that collects data from individuals to estimate the popularity of a movie among a certain population. The platform can learn the underlying distribution of privacy sensitivity through multiple interactions over time and other movies by using simpler mechanisms such as posted pricing. More specifically, consider a data acquisition mechanism that for each $i \in \mathcal{N}$ guarantees a local privacy level $\epsilon_i$ and central privacy level $\varepsilon/2$ and pays $t_i$, irrespective of the users' reported privacy sensitivities. With this mechanism, user $i$'s utility becomes*
>
> *$t\_i - c\_i \varepsilon\_i - (1-c\_i) \varepsilon\_i/2 = t\_i - \varepsilon\_i/2 - c\_i  \varepsilon\_i/2$*
>
> *which depends on $c_i$ only through the coefficient of $\varepsilon_i$. This mechanism is incentive compatible because the privacy levels and payments are not functions of the reported sensitivities. Moreover, user $i$ shares her data if the individual rationality constraint holds. This, in turn, implies that user $i$ shares her data if $c_i$ is below a certain threshold. Therefore, by varying the local privacy levels $\varepsilon_{i}$ the platform can learn the CDF of a user's privacy sensitivity from other users with similar characteristics (whose privacy sensitivity distribution is the same).
> Please note that removing the known prior assumption changes the problem (see, e.g., Bergemann and Morris (2012)), and we agree with you that an interesting future direction is to consider the prior-free variation of our setting.*
>
>
> **I am a little confused about the privacy sensitivities $(c_1, …, c_n)$. The authors mentioned that it should be unknown to the platform, however, in line 256, it is assumed that “the privacy sensitivity is $c_i$ and she reports $c’_i$”, also, in algorithm 1, the privacy sensitivities are also assumed to be given. Have the sensitivities been reported by the users? If not, how to estimate the privacy sensitivities when optimization?**
>
> *Thank you very much for your comment highlighting the need to explain this point better. We formulate the data acquisition problem in a Bayesian mechanism design framework. This requires the platform to know the distribution of sensitivities, as you mentioned above, and also the users to know the realization of their own sensitivities. However, the platform does not know users' realized sensitivity, and users may misreport those. Therefore, as you mentioned, the platform should design the optimal mechanism to incentivize users to report their sensitivities truthfully and simultaneously maximize the platform’s utility.*
>
> **In line 330, a Gaussian mechanism is given. What is the sensitivity L_i for the linear estimation of the sample mean?**
>
> *Here, $L_i$ is the weight of user $i$ in the estimator, which is equal to $w_i$.*

---

### Official Review · Reviewer_5KYG · 2022-07-11

**Rating:** 7
**Confidence:** 2
**Soundness:** 3 good
**Presentation:** 3 good
**Contribution:** 3 good

**Summary:**

This paper considers a heterogeneous population of users with respect
 to their privacy preferences. They consider the 'change'
 neighjbourhood for DP, which is natural for local DP. It first shows
 a lower bound on the linear estimation for the Gaussian-mechanism,
 and a matching upper bound.  linear estimator for individuals with
 $\epsilon_i$ preferences.  In the main paper, however, they consider
 relative privacy sensitivity to local and central mechanisms instead.

They then cast the case of MSE in parameter estimation, with incentive
compatibility and individual rationality constraints.
In some sense the result that it's optimal to only add local noise is
not extremely surprising, given the results about anonymity and local
DP.

**Questions:**

What can you say about generalisation to other estimators or non-MSE loss?

It is a bit strange that you initially look at the case of heterogenous overall losses, and then only tradeoff between local and central losses. Why is that?

The obvious question here is elicitation: How can users be sufficiently informed as to be able to truthfully report in the first place? I realise that this is a general question in mechanism design, but it is typically easier to price a tangible item than relative privacy losses...


**Limitations:**

Yes

**Strengths And Weaknesses:**

It is a well written and interesting combination of a number of areas that have not been studied very much so far, even though the design of optimal mechanisms for privacy is important.

 Under the assumption of a monotonic increasing 'virtual cost' (it's ok if you
don't explain it, but it is not actually fully defined: what are $f_i,
F_i$?)

---

> ### Author Response · Authors · 2022-08-02
> **Response to Reviewer 5KYG**
>
> *Thank you very much for your time in reviewing our paper and for providing valuable feedback. We are grateful for your positive assessment. Below, we respond to your comments in the order they appear in your report.*
>
> **Under the assumption of a monotonic increasing 'virtual cost' (it's ok if you don't explain it, but it is not actually fully defined: what are $f_i,F_i$?)**
>
> *Following your comment, in Assumption 1, we recall that $f_i(\cdot)$ and $F_i(\cdot)$ are probability density and cumulative distribution functions of $c_i$, respectively.*
>
> **What can you say about generalisation to other estimators or non-MSE loss?**
>
> *Thank you for this excellent question. We consider estimating the mean population (i.e., minimizing the mean squared error). However, our framework allows for considering other estimates as well. In particular, if we replace the mean square error in Equation (6) with the error of any other estimate, our Theorem 2 still holds: it converts the data acquisition mechanism design problem into an optimization problem. Therefore, the mechanism design problem boils down to solving a (potentially) non-convex optimization. Following your comment, we have highlighted this point in the conclusion.*
>
> **It is a bit strange that you initially look at the case of heterogenous overall losses, and then only tradeoff between local and central losses. Why is that?**
>
> *Thanks very much for highlighting the need to better clarify this point. We first design mechanisms to deliver heterogenous overall losses (both central and local) as a building block of our analysis. Let us explain this point further:
> We consider users that are heterogeneous in how they value local and central differential privacy, which is captured by $c_i$. In our data acquisition mechanism, we need to deliver heterogeneous local privacy losses and central privacy losses to users with different $c_i$. Now, we use the building block to deliver a given vector of local and central privacy losses.*
>
> **The obvious question here is elicitation: How can users be sufficiently informed as to be able to truthfully report in the first place? I realise that this is a general question in mechanism design, but it is typically easier to price a tangible item than relative privacy losses…**
>
> *Thanks very much for this stimulating question. We formulate the data acquisition problem in a Bayesian mechanism design framework. Therefore, as you mentioned, users need to know their ``type’’, which is adopted in Bayesian mechanism design pioneered by Myerson (1981). We agree that extending the analysis to a setting in which users do not have full information over their type in mechanism design and, more specifically in our setting, is a very interesting direction to explore. In a classical mechanism design setting, this question has been studied, among others, by Hartline and Lucier (2010), Hartline and Lucier (2015), and more recently by Balseiro et al. 2021.*

---

### Official Review · Reviewer_vRJn · 2022-07-11

**Rating:** 7
**Confidence:** 2
**Soundness:** 3 good
**Presentation:** 3 good
**Contribution:** 3 good

**Summary:**

This paper addresses an important problem in the field that is to eliminate local and central differential privacy (DP) concerns with one unified framework. It also considers the heterogeneous differential privacy where different users have different privacy needs. The approach is relatively simple compared to state-of-the-art solutions. They use Renyi DP and find a minimax lower bound first. Then they establish a point-wise optimization problem to characterize the optimal data acquisition mechanism.

**Questions:**

As far as I know, local DP is stronger and when it is guaranteed, central DP is also guaranteed. Is that correct?  If that's the case why you add noise twice in the framework?

**Limitations:**

Authors discusses the limitations in conclusion section. They mention that it is still an open problem to study the optimality of linear estimators. I don't have any other concern.

**Strengths And Weaknesses:**

Strengths: This paper addresses an important problem for real-world applications and I found the simplicity of the approach very practical for those applications. It is well written and theoretically sounds. Although I am not an expert in the field, I found many parts of the paper easy to follow and understand.

Weaknesses: This approach is quite applicable to real-world problems. I'd like to see some empirical evaluation of the proposed approach to understand how useful it could be for real applications. Minor but I've noticed a few typos, e.g. hose in page 9, line 362, We (capital W) in  page 2 line 87.

---

> ### Author Response · Authors · 2022-08-02
> **Response to Reviewer vRJn**
>
> *Thank you very much for your time in reviewing our paper and for providing valuable feedback. We are grateful for your positive assessment. Below, we respond to your comments in the order they appear in your report.*
>
> **This approach is quite applicable to real-world problems. I'd like to see some empirical evaluation of the proposed approach to understand how useful it could be for real applications. Minor but I've noticed a few typos, e.g. hose in page 9, line 362, We (capital W) in page 2 line 87.**
>
> *Thanks for mentioning the typos, which we have corrected. Thank you also for your comment, highlighting the need to better explain how our framework could potentially be used for real applications. We would like to mention that the idea of paying users for their data is considered in some real-world applications. For instance, there are companies such as Hu-manity.co that are aiming to pay users for their data, and in fact, they offer a spectrum of options based on the demanded level of privacy by users. More generally, the idea that users own their data and they should be paid to compensate for their data has been discussed in Economics literature as well (e.g., please see [a] for a book-length discussion).*
>
> *[a] Radical Markets: Uprooting Capitalism and Democracy for a Just Society, by Eric
> A. Posner and E. Glen Weyl, Princeton University Press, 2018.*
>
>
> **Questions:
> As far as I know, local DP is stronger and when it is guaranteed, central DP is also guaranteed. Is that correct? If that's the case why you add noise twice in the framework?**
>
> *Yes, as you mentioned, local DP implies central DP. In section 4.1 and in particular proposition 1, we show that it is optimal to have only local noises in the Gaussian mechanism, and therefore we start our model by adding only local noises. Please note that this is not ex-ante obvious. In particular, adding a noise centrally to the final estimator has an advantage because the weights in the final estimator give the platform a lever to deliver heterogeneous central privacy guarantees to users. Despite this advantage, we establish that adding noise centrally is never optimal. This is because the platform prefers to add the noise locally to contribute to both central and local privacy guarantees delivered to users.*

---

> > ### Comment · Reviewer_vRJn · 2022-08-08
> > **Response to rebuttal**
> >
> > Thanks to the authors for addressing the concerns. I read other reviews and the responses and I want to keep my score as is.

---

### Official Review · Reviewer_Buzh · 2022-07-11

**Rating:** 4
**Confidence:** 2
**Soundness:** 3 good
**Presentation:** 2 fair
**Contribution:** 2 fair

**Summary:**

This paper studies the mean estimation problem in the case that the users have heterogeneous sensitivities for local and central differential privacy. By sharing their data, the users receive payments, but lose both local and central privacy. A user’s goal is to maximize her total utility, while the platform’s goal is to minimize the sum of the mean squared error plus the total payments to all users with two constraints: incentive compatibility and individual rationality. The privacy loss is quantified by local Renyi DP and a generalized version of central Renyi DP, in which n different epsilons are given, one for each user. Each user’s “privacy sensitivity” is represented by a scalar $c_i$ in [0,1], which means that the weight of local DP loss for user i is $c_i$, and the weight of central DP loss is (1-$c_i$). The authors analyze the minimax lower bounds for the estimation error and propose a linear estimator that is order optimal. Then, they propose a data acquisition mechanism with the assumption that the server knows the agents’ privacy sensitivity distribution, and has both local and central privacy guarantees, and model it as a nonconvex optimization problem. Finally, they show that there is an approximately optimal polynomial time algorithm for the linear estimators.

**Questions:**

* Are there previous publications about users having different sensitivities for local and central DP, or different users having different central DP epsilons?
* There are some clarification questions about the setting / notations in “Strengths And Weaknesses” - “Quality”.
* Page 8, Proposition 1, it is claimed that it’s always optimal to add local noise. This is very counterintuitive, because usually local DP is much more expensive than central DP. What if every user only cares about central DP, but not local DP (all $c_i$ are 0). Does the conclusion still hold?


**Limitations:**

I don’t think the setting proposed in this paper is practical. See my comments in “Strengths And Weaknesses” - “Significance”.

**Strengths And Weaknesses:**

* Originality: This paper is related to central / local differential privacy and Bayesian data acquisition mechanisms. The authors propose a new privacy model, that each user has a privacy sensitivity, which determines whether they care more about central DP loss or local DP loss. Also, the users are allowed to have different epsilon values in both the local and central model. The problem is modeled by a nonconvex optimization problem, and approximate optimal polynomial time solutions are discussed. Based on the introduction and related work, the setting and solutions are novel. I do wonder if there are previous publications about users having different sensitivities for local and central DP, or different users having different central DP epsilons?

* Quality: The setting / notations are mostly ok. Some clarification questions:
  * Page 3, line 138, Is each $x_i$ a scalar? Or could it be a vector?
  * Page 3, line 141, Is $Z_i$ defined?
  * Page 3, line 142, “minimize the estimate’s error” What kind of estimate do you consider in this paper? Could it be arbitrary? From Page 4, line 173, seems you only consider estimating a scalar from user data?
  * Page 4, line 184, Is $\varepsilon_1^{(l)}$ defined before you use it?
  * Page 6, line 241 and 243, why is $\varepsilon_i^{(l)}: \mathcal{X} \rightarrow \hat{\mathcal{X}}$, but $\varepsilon_i^{(c)}: \mathbb{R_+^n} \rightarrow \mathbb{R_+}$? Shouldn’t both functions have the same input and output value range?

* Clarity: The writing and organization is mostly clear. Some comments:
  * Page 1, line 9, “The platform does not know the privacy sensitivity of users”. This might be misleading, because you do assume that the platform knows the distribution of users’ sensitivities (as stated in line 16). I suggest making the assumption clear in line 9.
  * Page 2, line 56, “be the more the second one” -> be more than the second one
  * Page 2, line 66, “The platform’s problem is to…” I suggest changing “problem” to “goal” or “target”. The same comment for other occurrences of this phrase.
  * Page 3, line 97, “in this literature” -> in the literature
  * Page 8, line 295-296, “This problem is still a functional optimization in terms …” -> This is still a functional optimization problem in terms …
  * Page 9, line 362, “a point-wise optimization problem hose…”, “hose” -> “whose”

* Significance: While the setting this paper proposes is new and kind of interesting, I wonder how practical it is to model privacy in a way that each user only has a “privacy sensitivity” in [0, 1] to determine the weight between central and local DP, and the platform got to determine every user’s epsilon value in both central and local DP. The idea is that a user is fine with arbitrarily large privacy loss, as long as they are paid according to some payment rule, which is hard to justify. If every user can have an upper bound epsilon, then the setting would be more meaningful.

---

> ### Author Response · Authors · 2022-08-02
> **Response to Reviewer Buzh**
>
> *Thank you very much for your time in reviewing our paper and for providing valuable feedback. Please find our responses below. In particular, we have clarified the novelty of our framework and analysis and highlighted our paper's practical motivation. We hope that the following addresses your questions.*
>
> **Comment: I do wonder if there are previous publications about users having different sensitivities for local and central DP, or different users having different central DP epsilons?**
>
> *We are unaware of any other work that considers users’ heterogeneous preferences for these two privacy measures. The existing papers focus on central (e.g., Gosh and Roth, 2011) and local differential privacy settings separately (e.g., Fallah et al., 2022).*
>
> **Comment: Page 3, line 138, is each $x_i$ a scalar? Or could it be a vector?**
>
> *In this paper, we focus on estimating a scalar mean by collecting users’ data. We agree that extending the results to vector data points is an interesting future direction. We would like to highlight that our framework is more general and allows for considering other potentially vector estimates (such as a linear estimator for vector data points). In particular, if we replace the mean square error in Equation (6) with the error of any other estimate, our Theorem 2 still holds: it converts the data acquisition mechanism design problem into an optimization problem. Therefore, the mechanism design problem boils down to solving a (potentially) non-convex optimization. Following your comment, we clarified this point in the conclusion.*
>
> **Comment: Page 3, line 141, Is $Z_i$ defined?**
>
> *We updated the paper to clarify that upper case letters denote random variables and lower case letters denote their realization.*
>
> **Page 3, line 142, “minimize the estimate’s error” What kind of estimate do you consider in this paper? Could it be arbitrary? From Page 4, line 173, seems you only consider estimating a scalar from user data?**
>
> *Our framework allows for considering other estimates (such as estimating higher moments of the distribution). In particular, if we replace the mean square error in Equation (6) with the error of any other estimate, our Theorem 2 still holds: it converts the data acquisition mechanism design problem into an optimization problem. Therefore, the mechanism design problem boils down to solving a (potentially) non-convex optimization. Following your comment, we clarified this point in the conclusion.*
>
> **Page 4, line 184, Is $\varepsilon_1^{(l)}$ defined before you use it?**
>
> *Please note that we are providing a mechanism that achieves any vector of local privacy losses in $\mathbb{R}_+^n$. Here, we are denoting the elements of this vector by a superscript (l) to indicate that they are local privacy losses.*
>
> **Page 6, line 241 and 243, why is $\varepsilon_i^{(l)}: \mathcal{X} \to \hat{\mathcal{X}}$, but $\varepsilon\_i^{(c)}:{\mathbb{R}\_{+}}^n \to \mathbb{R}\_{+}$? Shouldn’t both functions have the same input and output value range?**
>
> *Yes, we fixed $\varepsilon_i^{(l)}: \mathbb{R}\_+^n \to \mathbb{R}\_+$. We have also fixed other typos that you mentioned. Thank you very much for brining them to our attention.*
>
> **Significance: While the setting this paper proposes is new and kind of interesting, I wonder how practical it is to model privacy in a way that each user only has a “privacy sensitivity” in $[0, 1]$ to determine the weight between central and local DP, and the platform got to determine every user’s epsilon value in both central and local DP.**
>
> *Thanks for your comment highlighting the need to further motivate our setting. Our framework is motivated by the debate on the architecture of data markets. In particular, as highlighted by WILL.I.AM [2019] and GDMA [2018], some users prefer to protect their own raw data while others expect companies to protect their data proactively. Therefore, users have different preferences about local and central privacy. We model this preference by letting users weigh these two types of privacy losses differently. We then find the optimal data acquisition mechanism: in the optimal mechanism, the platform chooses the local privacy channels and the weights of users’ data. This, in turn, determines each user's local and central privacy loss.*
>
> *To more concretely answer your question, our current privacy loss in user $i$’s utility is the following function of the user’s preference and their local and central privacy losses:
> $c_i * \varepsilon_i^{(l)} + (1- c_i) * \varepsilon_i^{(c)}$*
>
> *Similar to the classical mechanism design, our framework is more general and can be extended to consider a general privacy loss which is a function of $c_i, \varepsilon_i^{(l)}$, and $\varepsilon_i^{(c)}$. However, solving for the optimal mechanism would depend on the functional form of the privacy loss and can potentially become even more challenging than our current non-convex optimization.*

---

> > ### Author Response · Authors · 2022-08-02
> > **Response to Reviewer Buzh (Continued)**
> >
> > **The idea is that a user is fine with arbitrarily large privacy loss, as long as they are paid according to some payment rule, which is hard to justify. If every user can have an upper bound epsilon, then the setting would be more meaningful.**
> >
> > *We would like to answer this question from two perspectives:*
> >
> > *First, we would like to highlight that the idea of paying users for their data is in fact considered in some real-world applications. For instance, there are companies such as Hu-manity.co that are aiming to pay users for their data, and in fact, they offer a spectrum of options based on the demanded level of privacy by users. More generally, the idea that users own their data and they should be paid to compensate for their data has been discussed in Economics literature as well (e.g., please see [a] for a book-length discussion).*
> >
> > *Second, the idea of asking users to provide their desired level of differential privacy loss may also have practical limitations. In particular, this would require users to understand differential privacy definitions well enough to be able to choose their desired $\varepsilon_i$. Note that, in our setting, in contrast, users do not choose their privacy losses. They only choose the ratio between their privacy loss for their interaction with the platform and their privacy loss for the interaction of the platform with the public.  Notably, the final privacy loss levels are endogenized through the mechanism. This would make the deployment of our mechanism easier.*
> >
> > *[a] Radical Markets: Uprooting Capitalism and Democracy for a Just Society, by Eric
> > A. Posner and E. Glen Weyl, Princeton University Press, 2018.*
> >
> >
> > **Are there previous publications about users having different sensitivities for local and central DP, or different users having different central DP epsilons?**
> >
> > *Regarding your first question, we are unaware of any other work that considers users’ heterogeneous preferences for these two privacy measures.*
> > *Regarding your second question,  Fallah et al., 2022 consider different users having different central DP epsilons. However, we differ from these works as one of our main contributions is providing a framework for studying the interplay between central and local privacy losses for users that weigh them differently.*
> >
> > **There are some clarification questions about the setting/notations in “Strengths And Weaknesses” - “Quality.”**
> >
> > *Thank you for your comments. We have updated the paper to address them.*
> >
> > **Page 8, Proposition 1, it is claimed that it’s always optimal to add local noise. This is very counterintuitive because usually, local DP is much more expensive than central DP. What if every user only cares about central DP but not local DP (all $C_i$s are $0$). Does the conclusion still hold?**
> >
> > *You are absolutely right that under the same level of privacy in central and local settings, local would be more costly in terms of the mean estimation error. However, please note that here we are not saying it is always optimal to replace a central DP noise with an equal local DP noise. In fact, the idea is that we replace the central noise with local noises with *a smaller* variance such that we keep the mean squared error and the central differential privacy parameters unchanged. However, since we move the noise to a local level, the local privacy losses decrease, and hence the total utility improves.
> > In other words, the main intuition is that if users care about privacy on both stages (central and local), then they would prefer to add a little noise locally, as it would boost privacy on both local and central levels simultaneously, rather than adding a central noise with a potentially higher variance which does not help with the local privacy loss.
> > For the special case that all $c_i=0$, i.e., users do not care about their local privacy loss, we can show that adding central noise leads to the same result as adding local noise. In other words, adding local noise is optimal, but it is not strictly optimal.*
> >
> >
> > **I don’t think the setting proposed in this paper is practical. See my comments in “Strengths And Weaknesses” - “Significance.”**
> >
> > *Thank you so much for pushing us to clarify the practical motivation of our paper. Please see above for the details. In summary: (i) we study the interplay between central and local privacy settings motivated by the users’ concern regarding the point of access to raw data and who should privatize users’ data, and (2) we incentivize users by payments by noting that the payment can either be in terms of ``free’’ services or direct payments adopted by some existing firms such as Hu-manity.co.*
> >
> >
> > *Once again, thank you for your valuable comments, and we would be happy to answer any further questions.*

---

> > > ### Comment · Reviewer_Buzh · 2022-08-09
> > > **Response to authors**
> > >
> > > Thank you for your response and revision!
> > >
> > > * For questions about whether $x_i$ is a scalar, the definition of $\varepsilon_1^{(l)}$ and other notations: my point is that you should define or clarify these notations the first time they appear or have a "preliminary" section for all notations in the paper.
> > > * About proposition 1: It is still confusing in the revised version that on line 329, you have “Despite this advantage, we establish that adding noise centrally is never optimal.” Also, I understand your logic that central DP could be achieved by adding a smaller local noise at each client, but this conclusion is not new and less interesting.
> > > * About the motivation of the setting: incentivizing users to contribute their data by paying a fee makes sense, but I’m still not comfortable with assigning potentially unbounded large epsilons to users, although it makes it easier to solve.

---

> > > > ### Author Response · Authors · 2022-08-09
> > > > **Response to Reviewer Buzh (second round)**
> > > >
> > > > Thank you very much for your response. We will clarify the notation per your instruction in the final version of our paper.
> > > >
> > > > Regarding your last comment on assigning potentially unbounded large epsilons:
> > > >
> > > > First, we would like to mention that the fact that we consider a setting with privacy sensitivities and payments is more a matter of modeling, and it is not just because it is easy to solve. In fact, this setting has been in the mechanism design and differential privacy literature since the pioneer work by [Ghosh & Roth, 2011].
> > > >
> > > > Furthermore, we would like to mention that the local epsilon (and hence central epsilon) assigned to a user in our setting is in fact bounded by a term decreasing in her privacy sensitivity.
> > > >
> > > > To see this, let us recall the optimization problem (7). Let us denote its optimal value by OPT. Setting $w_i=1/n$ and $y_i=1/n$, we achieve the following upper bound on OPT:
> > > >
> > > > $\text{OPT} \leq \frac{\sum_{i=1}^n \psi_i(c_i)}{n}+\frac{\gamma \alpha}{2} + \mathcal{O}(\frac{1}{n}),$
> > > >
> > > > which is a constant. Also, note that
> > > >
> > > > $(1-\psi_i(c_i)) \frac{w\_i^2}{\sum_{j=1}^n \frac{w\_j^2}{y\_j}} + \psi_i(c_i)y_i \quad (\star)$
> > > >
> > > > in (7) can be rewritten as
> > > >
> > > > $\frac{w\_i^2}{\sum_{j=1}^n \frac{w\_j^2}{y\_j}} + \psi_i(c_i) \left ( y_i - \frac{w\_i^2}{\sum_{j=1}^n \frac{w\_j^2}{y\_j}}\right ).$
> > > >
> > > > Hence, by conditioning on whether $\frac{w\_i^2}{\sum_{j=1}^n \frac{w\_j^2}{y\_j}}$ is greater than $y_i/2$ or not, we can establish that $(\star)$ is lower bounded by
> > > >
> > > > $\min(1,\psi_i(c_i)) \frac{y_i}{2}.$
> > > >
> > > > Therefore, we have the following upper bound on the optimal $y\_i^*$:
> > > >
> > > > $y\_i^* \leq \frac{2~\text{OPT}}{\min(1,\psi_i(c_i))}.$
> > > >
> > > > Finally, recall that we showed OPT is bounded by a constant. As a consequence, the privacy level assigned to a user who has non-zero privacy sensitivity will be bounded, and moreover, this bound decreases as her privacy sensitivity increases which is intuitive.
> > > >
> > > > We hope this derivation addresses your concern.

---

### Author Response · Authors · 2022-08-08
**General Comment**

We thank all the reviewers again for their comments and feedback. As the author-reviewer discussion period ends in less than 24 hours, please let us know if you have any further questions so that we can answer them in the remaining time.

---

### Meta-Review · Area_Chair_CnYN · 2022-08-25

**Recommendation:** Accept
**Confidence:** Certain

**Metareview:**

The reviewers all agreed that the problem setting is well-motivated, the proposed solution is novel, and that the paper is easy to understand and to follow. Though the authors' response addressed most of the reviewers' concerns, during the AC-reviewer discussion phase some reviewers still noted that:

* The method can add too much noise to data (vRjn);
* The setting where the server assigns different privacy levels ($\epsilon$) in exchange for payments may be unrealistic (Buzh);
* Prior knowledge of the distribution users' sensitivities may not be feasible in practice (BCdV).

Despite these issues remaining, the reviewers (and I) agree that the merits of the paper outweigh the relatively minor flaws above and that this paper would make a valuable addition to the conference. I encourage the authors to review the final version of their manuscript prior to publication.

**Award:**

No

---

### Decision · Program_Chairs · 2022-09-14

Accept